# Pruriception and neuronal coding in nociceptor subtypes in human and nonhuman primates

Amanda Klein[1,2], Hans Jürgen Solinski[3,4], Nathalie M Malewicz[5,6], Hada Fong-ha Ieong[5], Elizabeth I Sypek[7], Steven G Shimada[5], Timothy V Hartke[1], Matthew Wooten[1], Gang Wu[1], Xinzhong Dong[7,8], Mark A Hoon[3], Robert H LaMotte[5]*, Matthias Ringkamp[1]*

[1]Department of Neurosurgery, Neurosurgery Pain Research Institute, School of Medicine, Johns Hopkins University, Baltimore, United States; [2]Department of Pharmacy Practice and Pharmaceutical Sciences, University of Minnesota, Duluth, United States; [3]Molecular Genetics Section, National Institute of Dental and Craniofacial Research, Bethesda, United States; [4]Department of Experimental Pain Research, Medical Faculty Mannheim, University of Heidelberg, Mannheim, Germany; [5]Department of Anesthesiology, School of Medicine, Yale University, New Haven, United States; [6]Department of Anesthesiology, Intensive Care Medicine and Pain Management, Medical Faculty of Ruhr-University Bochum, BG University Hospital Bergmannsheil, Bochum, Germany; [7]The Solomon H. Snyder Department of Neuroscience, School of Medicine, Johns Hopkins University, Baltimore, United States; [8]Howard Hughes Medical Institute, Johns Hopkins University School of Medicine, Baltimore, United States

*For correspondence:
robert.lamotte@yale.edu (RHLM);
platelet@jhmi.edu (MR)

**Abstract** In humans, intradermal administration of β-alanine (ALA) and bovine adrenal medulla peptide 8–22 (BAM8-22) evokes the sensation of itch. Currently, it is unknown which human dorsal root ganglion (DRG) neurons express the receptors of these pruritogens, MRGPRD and MRGPRX1, respectively, and which cutaneous afferents these pruritogens activate in primate. In situ hybridization studies revealed that MRGPRD and MRGPRX1 are co-expressed in a subpopulation of TRPV1+ human DRG neurons. In electrophysiological recordings in nonhuman primates (*Macaca nemestrina*), subtypes of polymodal C-fiber nociceptors are preferentially activated by ALA and BAM8-22, with significant overlap. When pruritogens ALA, BAM8-22, and histamine, which activate different subclasses of C-fiber afferents, are administered in combination, human volunteers report itch and nociceptive sensations similar to those induced by a single pruritogen. Our results provide evidence for differences in pruriceptive processing between primates and rodents, and do not support the spatial contrast theory of coding of itch and pain.

## Introduction

The sensations of itch and pain serve a similar purpose, which is to alert the organism of potentially harmful external threats. Although the sensations of itch and pain can elicit different behavioral responses (such as scratching vs. withdrawal, rubbing or guarding), they are closely linked as they appear to be elicited by activity in one and the same type of fiber or functionally similar primary afferents.

In mice, pruritogens activate at least three populations of neurons (*Liu et al., 2012*; *Liu et al., 2009*; *Solinski et al., 2019b*). Among these, two neuronal populations express Mas-related

G-protein-coupled receptors (Mrgpr), namely *Mrgprc* and *Mrgprd*, in a non-overlapping fashion (*Zylka et al., 2003*), and intradermal administration of their respective agonist, bovine adrenal medulla peptide 8–22 (BAM8-22) and β-alanine (ALA), results in scratching behavior. The third population expresses natriuretic polypeptide b (*Nppb*) and mediates mast cell-induced itch. Neurons expressing *Mrgprc* or *Nppb* also express the histamine (HIS) receptor *Hrh1* and likely mediate scratching behavior induced by HIS.

In humans, HIS and the nonhistaminergic pruritogens ALA and BAM8-22 each elicit itch and nociceptive sensations (*Liu et al., 2012*; *Sikand et al., 2011a*; *Sikand et al., 2009*; *Sikand et al., 2011b*). HIS preferentially activates mechanically insensitive C-fiber afferents (C-MIAs) in human (*Schmelz et al., 1997*) and nonhuman primates (*Wooten et al., 2014*). In nonhuman primates, ALA preferentially excites QCs, a subtype of mechanoheat-sensitive, polymodal C-fiber (CMH) nociceptor that adapts quickly to noxious heat as opposed to those that adapt slowly (SCs), and C-MIAs are unresponsive to ALA (*Wooten et al., 2014*). The type of afferent nerve fiber activated by BAM8-22, an agonist of MRGPRX1 in primate (*Lembo et al., 2002*), is currently unclear. Based on the non-overlapping expression of *Mrgprd* and *Mrgprc* in mouse dorsal root ganglion (DRG) neurons and the preferential activation of QC fibers by ALA in nonhuman primate, one might expect that in primate MRGPRD and MRGPRX1 would be similarly expressed in non-overlapping DRG neurons and that ALA and BAM8-22 would activate different neuronal afferent populations.

In humans and other primates, multiple types of nociceptive primary afferents including unmyelinated and myelinated nerve fibers appear to have a role in itch sensation (*Johanek et al., 2008*; *Johanek et al., 2007*; *Namer et al., 2008*; *Ringkamp et al., 2011*; *Schmelz et al., 1997*; *Wooten et al., 2014*). Moreover, HIS and nonhistaminergic pruritogens activate distinct sets of nociceptive primary afferents, including mechano-insensitive C fibers (C-MIAs), CMHs, mechanosensitive A-fiber nociceptors (A-MSAs), and spinothalamic projection neurons (*Davidson et al., 2007*; *Johanek et al., 2008*; *Namer et al., 2008*; *Schmelz et al., 1997*; *Wooten et al., 2014*). The signaling of pruritic stimuli by nociceptive afferents that are also thought to encode nociceptive mechanical, thermal, and chemical stimuli (*Ringkamp et al., 2013*) poses a conundrum regarding the neuronal coding of these distinct sensations in the nervous system. Among the models that have been proposed to explain this conundrum (for review, see *Carstens et al., 2020*; *LaMotte et al., 2014*), the spatial contrast model posits that neuronal activity from a spatially restricted set of nociceptive afferents is interpreted as 'itch,' whereas activity from a spatially larger pool of afferents is interpreted as 'pain' (*Namer et al., 2008*; *Namer and Reeh, 2013*; *Steinhoff et al., 2019*). However, it has not been tested in humans whether a combination of pruritogens that activates a greater number and variety of primary afferents and thereby decreases the spatial contrast between activated and silent fibers could lead to reduced itch and/or increased nociceptive sensation.

Here, we probed the relationship between the expression of receptors for ALA and BAM8-22 in postmortem human DRGs and the physiological responses of these pruritogens in single cutaneous nerve fibers of nonhuman primate. We found that human DRG neurons co-express both, MRGPRD and MRGPRX1, and also transient receptor potential channels from the vanilloid subfamily V member 1 (TRPV1). Further, injection of ALA and BAM8-22 into the receptive fields (RFs) of nociceptors in nonhuman primate revealed that QCs respond to ALA, and SCs preferentially respond to BAM8-22, with a large degree of overlap. In accompanying psychophysical studies, we studied the effects of individual pruritogens and their combinations and recorded the reported itch, nociceptive sensations, and dysesthesias in human subjects. Since we know at least three pruritogens that can preferentially activate QCs, SCs, and C-MIAs, we additionally tested the hypothesis that the co-activation of the different sets of primary afferents by ALA, BAM8-22, and HIS would enhance itch and/or nociceptive sensations and dysesthesias in humans. In addition, in primates, HIS and cowhage activate largely non-overlapping spinothalamic tract (STT) neurons (*Davidson et al., 2012*). Therefore, by combining ALA, BAM8-22, and HIS, we tested psychophysically whether activating both types of STT neurons would result in altered itch or nociceptive sensations.

# Results

## MRGPRX1 and MRGPRD have overlapping expression patterns in human DRG

The representation of the MRGPR families of receptors in humans and nonhuman primates differs from rodents in that they contain a distinct MRGPRX subfamily of receptors that is absent in rodents and that they do not have either the rodent *Mrgpra* or *Mrgprc* subfamilies of receptors (*Figure 1A*; *Solinski et al., 2014*). Whether MRGPRX1 and MRGPRD are expressed in non-overlapping DRG neurons in primate, similar to what has been previously reported in mouse for *Mrgprc* and *Mrgprd*

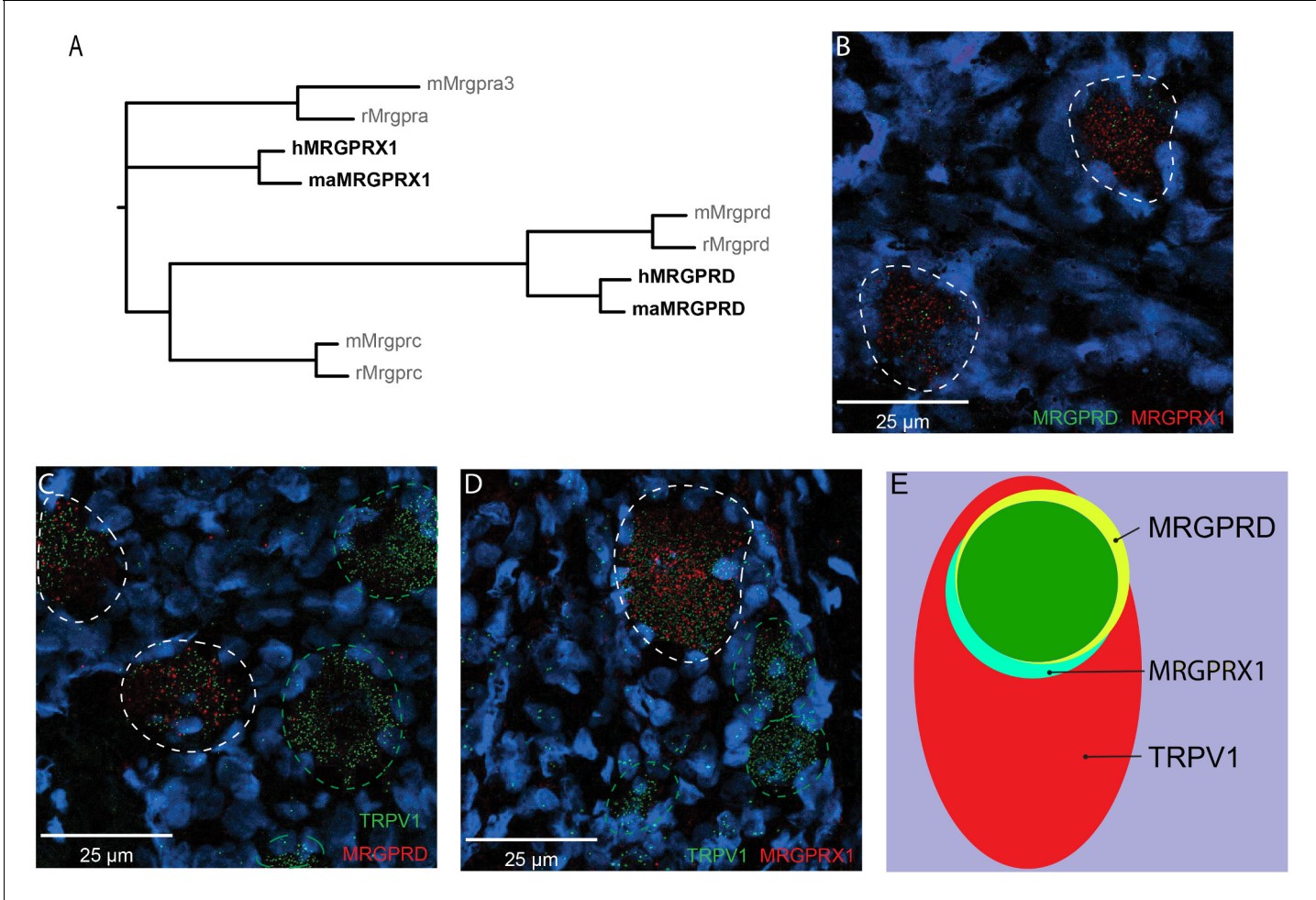

**Figure 1.** MRGPRX1 and MRGPRD are co-expressed in TRPV1-expressing human dorsal root ganglion (DRG) neurons. (**A**) Phylogenetic tree of MRGPRs from mouse (m), rat (r), macaque (ma), and human (h). Note that the MRGPRD gene is conserved among rodents and primates, while the Mrgpra and Mrgprc subfamilies are rodent-specific and the MRGPRX subfamily is primate-specific. For clarity, only one murine Mrgpra gene and only one macaque or human MRGPRX gene is shown. (**B–D**) Representative double in situ hybridization (ISH) images of a field of human DRG with neurons stained for MRGPRD (**B**, green; **C**, red), MRGPRX1 (**B** and **D**, red), and TRPV1 (**C** and **D**, green). Double-positive and single-positive neurons are outlined with white and green dashed lines, respectively. DAPI counterstain is displayed in blue. (**E**) Venn diagram summarizing the relative expression overlap of MRGPRD (yellow), MRGPRX1 (blue), and TRPV1 (red) in human DRG. Note that MRGPRD and MRGPRX1 are expressed in a largely overlapping population (green, i.e., MRGPRD + MRGPRX1/MRGPRD = 89.6 ± 1.5%; MRGPRX1 + MRGPRD/MRGPRX1 = 93.9 ± 2.0%) in about 1/3 of all TRPV1-positive neurons (MRGPRD + TRPV1/TRPV1 = 36.5 ± 4.2%; MRGPRX1 + TRPV1/TRPV1 = 32.5 ± 3.7%). Expression analysis for all three markers was performed in DRG tissue from four individuals, and data are stated as mean ± standard error of the mean (SEM). Green filled area indicates the overlap in expression of MRGPRD and MRGPRX1.

The online version of this article includes the following source data and figure supplement(s) for figure 1:

**Source data 1.** Expression of MRGPRX1 and MRGPRD in nonhuman primate DRG.

**Figure supplement 1.** MRGPRX1 and MRGPRD are co-expressed in macaque dorsal root ganglion (DRG) neurons.

(*Zylka et al., 2003*), is currently unclear. We therefore investigated the expression of MRGPRX1 and MRGPRD in human DRG using double-label in situ hybridization (ISH). We found that MRGPRX1 and MRGPRD were co-expressed in the same neurons (*Figure 1B*). Specifically, on average 89.6 ± 1.5% of DRG neurons positive for MRGPRD in a given donor were also positive for MRGPRX1 (380/426 neurons from four donors combined), and 93.9 ± 2.0% (380/407) of DRG neurons positive for MRGPRX1 were also positive for MRGPRD. Previously it has been reported in a *Mrgprd* reporter mouse line that *Mrgprd+* neurons were TRPV1 negative and non-peptidergic CMHs (*Rau et al., 2009*). If findings in human DRG were comparable to mouse, then one would expect to observe no overlap between TRPV1 and MRPGRD expression. However, we found that MRGPRD neurons nearly always co-expressed TRPV1 (*Figure 1C*) with 95.2 ± 1.6% (479/510) of MRGPRD+ neurons being also positive for TRPV1. In the mouse, *Mrgprc* is expressed in a subset of capsaicin-responsive neurons (*Liu et al., 2009*). Recent ISH data from human DRG show that all MRGPRX1+ neurons co-express NPPB, and that NPPB-expressing neurons are a subset of TRPV1+ neurons (*Solinski et al., 2019a*). In agreement with these previous findings, we observed that 100% (429/429) of MRGPRX1+ neurons were also positive for TRPV1 (*Figure 1D*). Of TRPV1+ neurons, 32.5 ± 3.7% (429/1273) and 36.5 ± 4.2% (479/1211) of cells also expressed MRGPRX1 and MRGPRD, respectively (*Figure 1E*). In parallel with these studies, we characterized the expression of MRGPRD and MRGPRX1 in DRG of macaques. We also found, in this primate, that these two receptors are expressed in a largely overlapping fashion, although with slight differences (*Figure 1—figure supplement 1*). More specifically, of MRGPRD+ neurons, 78.1 ± 6.1% (199/238) co-express MRGPRX1 while of MRGPRX1+ neurons, only 57.1 ± 11.6% (199/448) co-express MRGPRD (*Supplementary file 1*). Together our results point to profound species-dependent differences of MRGPR expression patterns between rodents and primates with additional slight differences inside the primate lineage.

## BAM8-22 preferentially activates SCs in nonhuman primates

To learn more about the neurons that are activated by ALA and BAM8-22, we probed the different functional classes of neurons using teased-fiber recordings from single nerve fibers in the primate *Macaca nemestrina* (*Wooten et al., 2014*). Across all CMHs tested, the average number of action potentials (APs) elicited by BAM8-22 was significantly larger than the response to ALA (74.95 ± 8.88 APs vs. 47.44 ± 6.05 APs, respectively; paired t-test: $t_{(65)} = -2.142$, p=0.0359; *Figure 2A*). Of the 66 CMHs tested, 1 fiber was unresponsive to either agonist, 26 responded only to BAM8-22, 11 only to ALA, and 28 to both agonists (*Figure 2B*). Thus, BAM8-22 activated 54/66 CMHs, whereas ALA activated 39/66 fibers. Across all fibers tested, the incidence of responsiveness to BAM8-22 and ALA was significantly different ($X^2_{(1)}$=6.44, p=0.0112). The response profiles for ALA and BAM8-22 across the responsive CMHs were indistinguishable, except for the first 10 s following the injection when BAM8-22-induced activity was about twice that of ALA. The duration of action potential activity was similar after BAM8-22 and ALA injection (*Figure 2C*).

Based on responses to a stepped heat stimulus (49℃, 3 s, *Figure 3A*), CMHs can be classified into two distinct subtypes, QCs and SCs (*Wooten et al., 2014*). Briefly, heat responses of QCs exhibit a burst of discharge at the onset of the heat stimulus with the discharge adapting during the stimulus plateau. SCs typically have a slower response onset, do not exhibit a burst at the onset of the stimulus, and the peak discharge occurs during the plateau phase of the stimulus (*Figure 3B*). QCs and SCs can be formally separated by plotting the time of peak discharge (relative to stimulus onset) against the sum of temperature rise time plus the minimum conduction latency from skin (*Figure 3C*). For 31/66 CMHs, the time of peak discharge fell above the line of equality (*Figure 3C*, black line), indicating that the peak discharge occurred during the plateau phase of the stimulus, and these afferents were therefore classified as 'SC' (*Figure 3C*, blue circles). 29 out of 66 CMH fibers exhibited a burst of discharge at the onset and then an adapting response, and these afferents were classified as 'QC' (*Figure 3C*, red circles). For 26/29 QCs, the time of peak discharge fell below or close to the line of equality, indicating occurrence of peak discharge during or at the end of the rising phase of the temperature ramp. For the remaining three QC fibers, the time of peak discharge fell above the line of equality. Another six CMHs, in which the time of peak discharge fell close to the line of equality and which did not exhibit an initial burst discharge typical of a QC response, were labeled as 'unclassified' (*Figure 3C*, gray circles). The filled red and blue circles represent the data points for the specimen heat responses shown in *Figure 3B*.

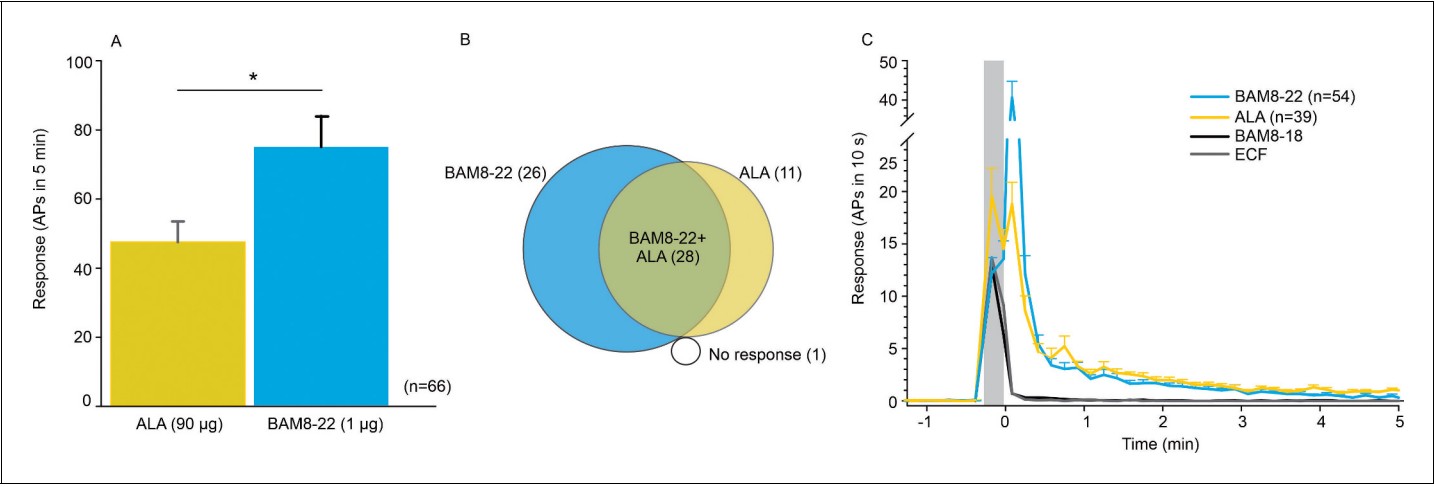

**Figure 2.** CMHs respond to bovine adrenal medulla peptide 8–22 (BAM8-22) more vigorously than to β-alanine (ALA) but with a similar time course. (**A**) The average evoked response of all CMHs to BAM8-22 (blue) was significantly larger than that to ALA (yellow) injection (paired t-test: $t_{(65)}$ = −2.142, p=0.0359). (**B**) Venn diagram of the number of CMHs responsive to BAM8-22 or ALA or both. (**C**) The time course of action potential activity (plotted as number of action potentials in 10 s bins) was similar after BAM8-22 and ALA injection, except for the greater response to BAM8-22 within the first 10 s following injection. The average responses of the same populations to vehicle (extracellular fluid [ECF] and BAM8-18) are graphed with gray and black lines, respectively. Error bars represent standard error of the mean (SEM). Gray box marks time of needle insertion and injection.

The online version of this article includes the following source data for figure 2:

**Source data 1.** Activation of CMHs by ALA and BAM8-22.

We found that the two subpopulations for CMHs, QCs and SCs, differed in their responses to injections of ALA and BAM8-22 into their cutaneous RFs. The differences are shown first by the responses of a typical QC and SC fiber in *Figure 3D, E* and summarized for the two populations in *Figure 4*. For the specimen QC (*Figure 3D*, same fiber for which the heat response is plotted in *Figure 3B*), injection of ALA (top panel) and BAM8-22 (bottom panel) induced a neuronal response for about 5 min, and the response to ALA (81 APs) was larger than to BAM8-22 (50 APs). Vehicle (ECF, BAM8-18) did not produce any activity in this afferent beyond the injection (data not shown). In the SC-fiber (for which the heat response is shown in *Figure 3B*), ALA administration did not result in activity outlasting the injection (*Figure 3E*, top panel) and the preceding ECF injection (response not shown) produced a total response of 18 APs. In contrast, injection of BAM8-22 (*Figure 3E*, bottom panel) produced a vigorous response of 198 APs within 5 min. The preceding BAM8-18 injection only produced activity during the injection (response not shown). Time courses of ALA- and BAM8-22-induced activity across the QC- and SC-populations are shown in *Figure 3F and G*, respectively. In QCs and SCs, vehicles did not evoke activity beyond the injection period. In QCs, ALA and BAM8-22 caused excitation with a similar time course (*Figure 3F*). For both agonists, the highest activity was observed immediately following the injection, and the activity decreased throughout the 5 min observation period. At 5 min post injection, the number of APs in QCs over 10 s was 0.55 ± 0.29 and 1.14 ± 0.23 for BAM8-22 and ALA, respectively. In contrast, in SCs, only BAM8-22-induced activity outlasted the injection and was different from the response to vehicle injections (*Figure 3G*). The highest activity occurred during the first 10 s following the injection, and the response decreased within approximately 3 min to a level similarly seen following vehicle injection.

To investigate further whether responses to ALA and BAM8-22 differed between QCs and SCs, the net response to BAM8-22 was plotted against that to ALA for each individual afferent (*Figure 4A*). For 20/29 QCs, data points fell below the diagonal line of equal responsiveness, indicating that responses to ALA were larger than responses to BAM8-22. In contrast, for 26/31 SCs, responses to BAM8-22 were greater than those to ALA. In fact, ALA responses in only seven SCs fulfilled the criterion of a positive response (≥10 APs). Responses to ALA and BAM8-22 differed significantly within and across fiber types (*Figure 4B*, repeated measures ANOVA (RMANOVA) with' fiber type' as a between-subjects factor and' pruritogen' as within-subjects factor; interaction: $F_{(1,58)}$=29.55, p<0.001). Post hoc analysis revealed that for SCs (n = 31), responses to BAM8-22 were

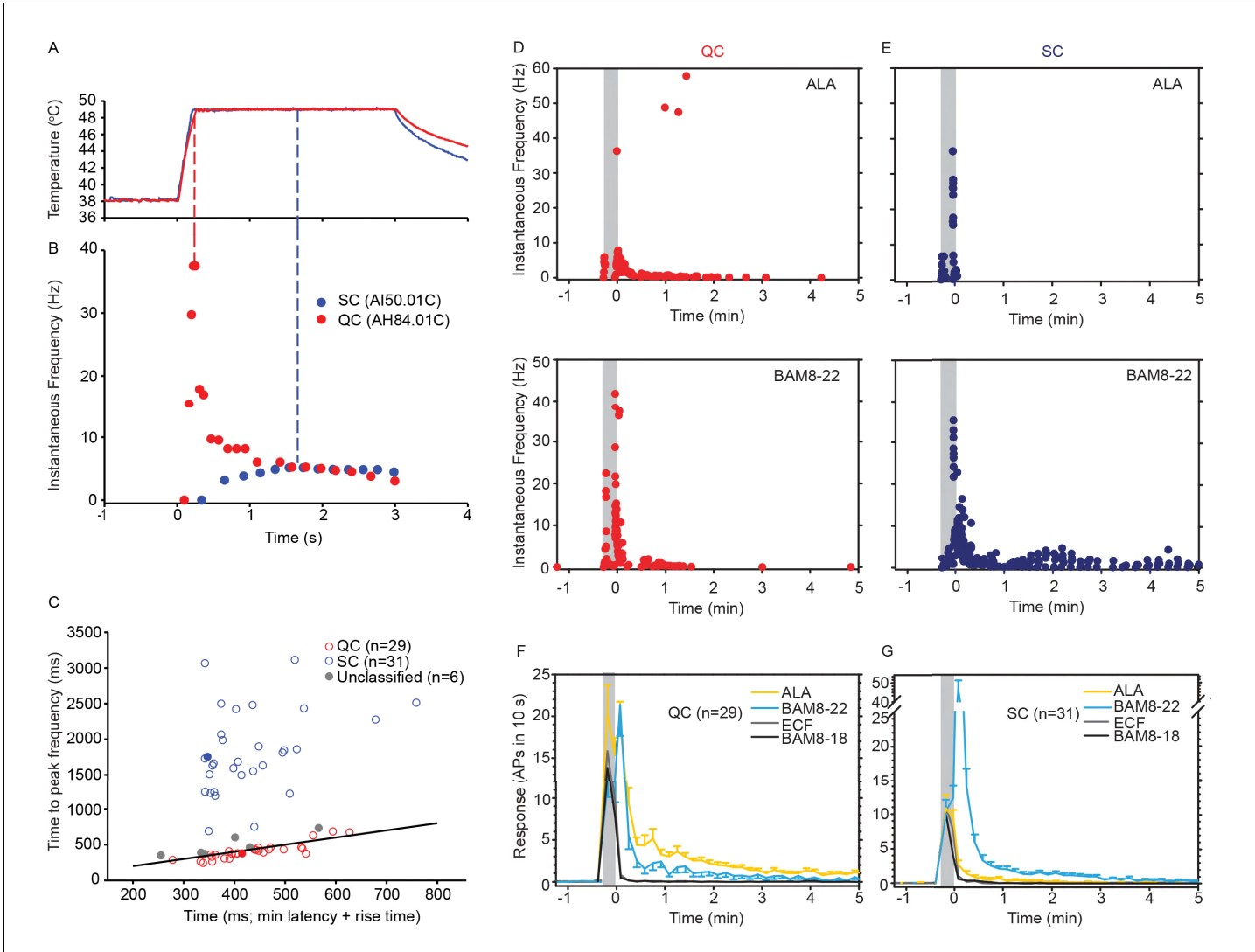

**Figure 3.** CMH subpopulations, QCs and SCs, exhibit differences in their response to heat stimulation and in the magnitude and time course of their response to β-alanine (ALA) and bovine adrenal medulla peptide 8–22 (BAM8-22). (**A**) Temperature waveform of the $CO_2$ laser-evoked heat stimulus. The skin was first pre-heated to 38 °C, 3 s baseline temperature and then rapidly raised (rise time ~200 ms) to 49 °C for 3 s. (**B**) Specimen recording showing the response of an individual QC fiber (red circles) and an SC fiber (blue circles) to the laser-heat stimulus described in (**A**). The instantaneous discharge frequency is plotted versus time. Each dot represents the occurrence of an action potential (AP). The response in the QC fiber starts during the temperature rise and reaches peak frequency at the end of the ramp (red dashed line). The SC fiber reached the peak instantaneous discharge frequency during the plateau phase of the heat stimulus (blue dashed line). (**C**) The time of peak discharge for each fiber is plotted against the sum of stimulus rise time + the minimal AP conduction time as measured in response to transcutaneous electrical stimulation from the proximal edge of the receptive field (RF). Data points above the line of equality correspond to fibers in which the peak discharge occurred in the plateau phase of the heat stimulus (SCs, blue circles). Data points falling below the line of equality are from those fibers whose peak discharge occurred during the rising phase of the heat stimulus (QCs, red circles). The filled circles represent the data from the specimen recordings shown in (**B**) and also of the specimen responses to ALA and BAM8-22 shown in (**D**) and (**E**). Gray circles indicate data from fibers that were unclassified. Examples of responses in (**D**) of a QC fiber and (**E**) an SC fiber to ALA (top panels) and each to BAM8-22 (bottom panels). Responses to ALA and BAM8-22 are from the same fiber. The instantaneous frequency of each AP is plotted versus the time of its occurrence. The time course of neuronal activity induced by ALA, BAM8-22, and vehicle controls in the population of (**F**) QC fibers and (**G**) SC fibers. The average number of APs recorded over 10 s intervals during the 5 min observation period following injection is plotted. In QC fibers, ALA and BAM8-22 produced marked excitation, whereas in SC fibers, only BAM8-22 produced long-lasting activity. Error bars represent standard error of the mean (± SEM). Gray boxes mark time of needle insertion and injection. The online version of this article includes the following source data for figure 3:

**Source data 1.** Activation of QCs and SCs by ALA and BAM8-22.

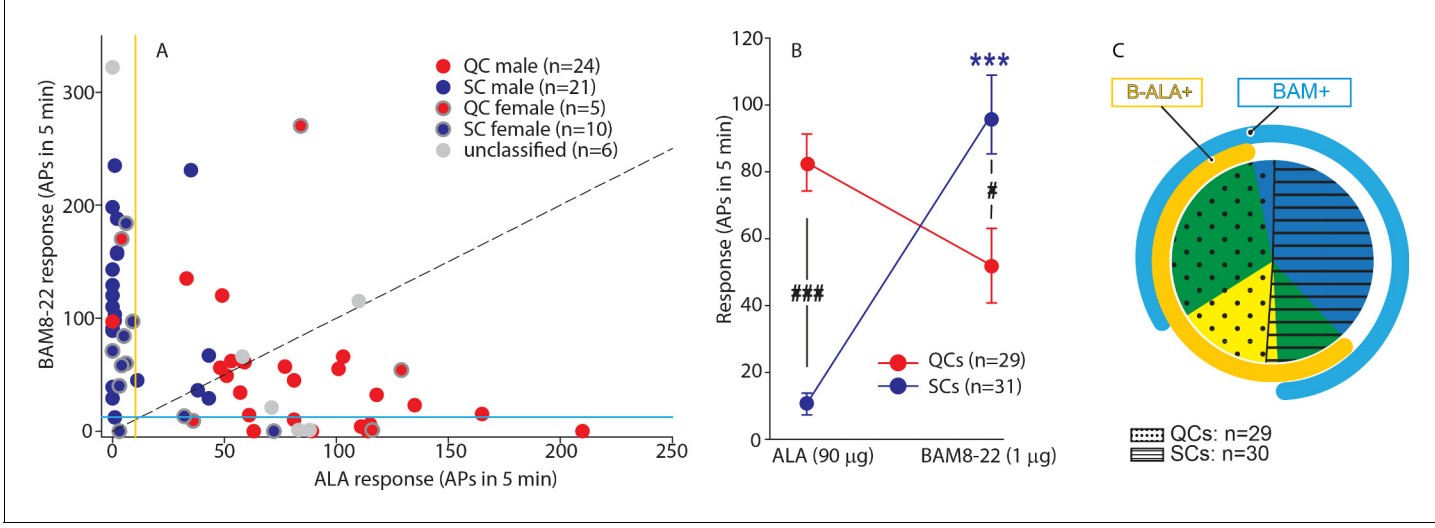

**Figure 4.** QCs and SCs from both male and female monkeys differ in sensitivity and incidence of activation by β-alanine (ALA) and bovine adrenal medulla peptide 8–22 (BAM8-22). (**A**) For each C-fiber, the response to BAM8-22 was plotted against its response to ALA. Solid red and blue circles indicate QC and SC fibers, respectively, recorded from male animals. Red and blue circles with a gray border indicate data obtained from female animals. Solid gray circles indicate data from C fibers with an unclassified heat response. The diagonal line indicates equal response to both compounds. Vertical and horizontal lines indicate the 'threshold' (≥10 action potentials [APs]) for an afferent to be counted as being responsive to an agonist (yellow and blue, respectively, for responses to ALA and BAM8-22). (**B**) Population responses of QCs and SCs to ALA and BAM8-22. In SCs, intradermal injection of BAM8-22 produces a significantly greater response than ALA (*** p<0.0001). The response to ALA was significantly larger in QCs than SCs (###p<0.0001), whereas the response to BAM8-22 was significantly larger in SCs than QCs (#p=0.002). Data were analyzed with repeated measures ANOVA (RMANOVA) with 'fiber type' as between-subjects factor and 'pruritogen' as a within-subjects factor ('fiber type' × 'pruritogen': $F_{(1,58)}$=29,55; p<0.0001), followed by Scheffe test for post hoc analysis. (**C**) ALA activated 27/29 QCs and 7/31 SCs. BAM8-22 activated 21/29 QCs and 29/31 SCs. The occurrence of QCs and SCs responding to ALA only (eight and one afferents, respectively) was fairly rare. The number of QCs responding only to BAM8-22 (two units) was also small, whereas the majority of SCs (23/31) only responded to BAM8-22. Of the 31 SCs, one did not respond to either agonist. 19 QCs and 6 SCs responded to both, ALA and BAM8-22.

The online version of this article includes the following source data for figure 4:

**Source data 1.** Preferential activation of QCs and SCs by ALA and BAM8-22.

significantly larger than those to ALA (94.00 ± 12.01 APs vs. 10.32 ± 3.22 APs, p<0.001, Scheffe test). In QCs (n = 29), responses to ALA did not significantly differ from those to BAM8-22 (83.38 ± 8.42 APs vs. 51.97 ± 11.17 APs, respectively; p=0.25). Between populations, responses to ALA were significantly larger in QCs than SCs (p<0.001, Scheffe test), whereas responses to BAM8-22 were significantly larger in SCs than QCs (p=0.021, Scheffe test). ALA activated 27/29 QCs and 19 of these responded also to BAM8-22 (*Figure 4C*). Two QCs were only responsive to BAM8-22. In contrast, only 7/31 SCs responded to ALA with 6 of these being also responsive to BAM8-22, whereas 23/31 SCs responded only to BAM8-22. One of 31 SCs did not respond to either agonist. Of the six unclassified afferents, three units responded to both agonists, two responded to ALA only, and the remaining only to BAM8-22. Taken together, these findings suggest that, at the doses tested, QCs and SCs are preferentially activated by ALA and BAM8-22, respectively. The data imply that QCs may represent fibers expressing both MRGPRs at the peripheral terminals, but preferentially MRGPRD, whereas SCs may preferentially express MRGPRX1. To summarize, in contrast to our human ISH data that suggested nearly 100% overlap of MRGPRX1 and MRGPRD expression, we found that only about half of the CMH neurons tested responded to both BAM8-22 and ALA. Focusing more specifically on macaques, our combined ISH and electrophysiology data suggest that some CMHs (i.e., QCs) express both MRGPRs, while others (i.e., SCs) are functionally preferentially activated by BAM8-22 and only express MRGPRX1. Furthermore, in accordance with MRGPRD and MRGPRX1 neurons expressing TRPV1, all of these mechanosensitive C-fibers responded to noxious heat regardless of the selectivity of their responses to BAM8-22 or ALA.

## C-MIAs and A-fiber nociceptors are less frequently activated or unresponsive to ALA and BAM8-22

We also investigated whether BAM8-22 and ALA activate C-MIAs and A-fiber nociceptors that are involved in mediating itch sensation to HIS (*Schmelz et al., 1997*; *Wooten et al., 2014*) and cowhage (*Ringkamp et al., 2011*), respectively. We previously reported that seven C-MIAs were tested with ALA and that none responded (*Wooten et al., 2014*). Of six C-MIAs tested with BAM8-22 in this study, all were tested with heat and four responded. Two of six C-MIAs (one responsive to heat) responded to BAM8-22. One of the C-MIAs had a 5 min response of 146 APs with 112 APs occurring during the first 30 s following the injection, and the response ending within 5 min. The other C-MIA had a 5 min response of 122 APs, of which 50 APs occurred during the first 30 s after injection. The response lasted for 20 min with a total response of 171 APs. We tested 16 mechanosensitive and 8 mechanoinsensitive A-fiber nociceptors (A-MSA and A-MIAs, respectively). Of the 16 A-MSAs, one responded to ALA (5 min net response of 14 APs), and another to BAM8-22 with 129 APs within the first 30 s after injection but no activity thereafter. Of the eight A-MIAs, none responded to ALA or BAM8-22. These data suggest that A-fiber nociceptors and C-MIAs are likely minor contributors to the sensations caused by these compounds.

## In humans, a combination of pruritogens does not change the itch, nociceptive sensations, dysesthesias, or skin reactions compared to the effects of one of the component pruritogens given alone

In humans, cowhage-induced itch and nociceptive sensations are mediated by activation of nociceptive unmyelinated and myelinated afferents that are also thought to mediate pain from noxious heat and mechanical stimuli (*Johanek et al., 2008*; *Namer et al., 2008*; *Ringkamp et al., 2013*; *Ringkamp et al., 2011*). How the sensations of itch and pain are mediated by activity in the same set of afferent nerve fibers is currently unclear, and several models have been proposed to solve this puzzle (*Carstens et al., 2020*; *LaMotte et al., 2014*). Among these, the 'spatial contrast' model posits that locally restricted activation of a small population of pruriceptive nociceptors induces itch sensation, while activation of a broader and greater number of nociceptors induces the sensation of pain (*Namer et al., 2008*; *Namer and Reeh, 2013*; *Steinhoff et al., 2019*). Our present electrophysiological results indicate that ALA and BAM8-22 preferentially activate different sets of cutaneous CMHs in monkey (*Figure 4*). Previous studies have shown that HIS-induced itch is likely mediated by C-MIAs (*Schmelz et al., 1997*; *Wooten et al., 2014*). Therefore, the 'spatial contrast' model can be tested experimentally by concurrent intradermal injections of combinations of ALA, BAM8-22, and HIS and comparison of the induced psychophysical responses to those from injection of an individual pruritogen.

The majority of subjects reported itch and nociceptive sensations after the administration of a single pruritogen and after the combinations thereof. The number of subjects reporting these sensations did not differ between stimuli (*Figure 5A*), and the number of subjects experiencing itch or nociceptive sensations did not change when combinations of pruritogens were applied. For the co-administration of BAM8-22 and ALA, the temporal profiles of the different sensations were visibly similar compared to those evoked by BAM8-22 or ALA alone, peaking within the first minute after application and decreasing slowly over approximately 10 min (*Figure 5B–D*). Similarly, HIS-induced itch peaked within the first minute after injection but appeared to decline much slower over 20 min (*Figure 5E*). The temporal profile of the itch sensation produced by the triple combination was similar to that of the BAM8-22 + ALA combination, except that the decline of itch sensation appeared to be slower (*Figure 5E*). The time courses of the nociceptive sensations following HIS were similar to those of the BAM8-22 + ALA combination and the combined application of all three pruritogens (*Figure 5F, G*).

To quantify and compare the responses to each of the five stimuli (three single pruritogens and two combinations), the following three measurements were obtained: the peak magnitude rating of perceived intensity, the area under the rating curve (AUC), and the duration of the ratings. For each of these measurements, differences in mean response were initially statistically analyzed with a 'stimulus' (5) × 'sensory quality' (3) × 'sex' (2) RMANOVA. Only for peak ratings, but not for AUC and duration,' sex' had a significant effect, with females giving significantly larger peak ratings than males ($F_{(1,27)}$=6.53, p<0.017). However, none of the interactions with the other factors reached

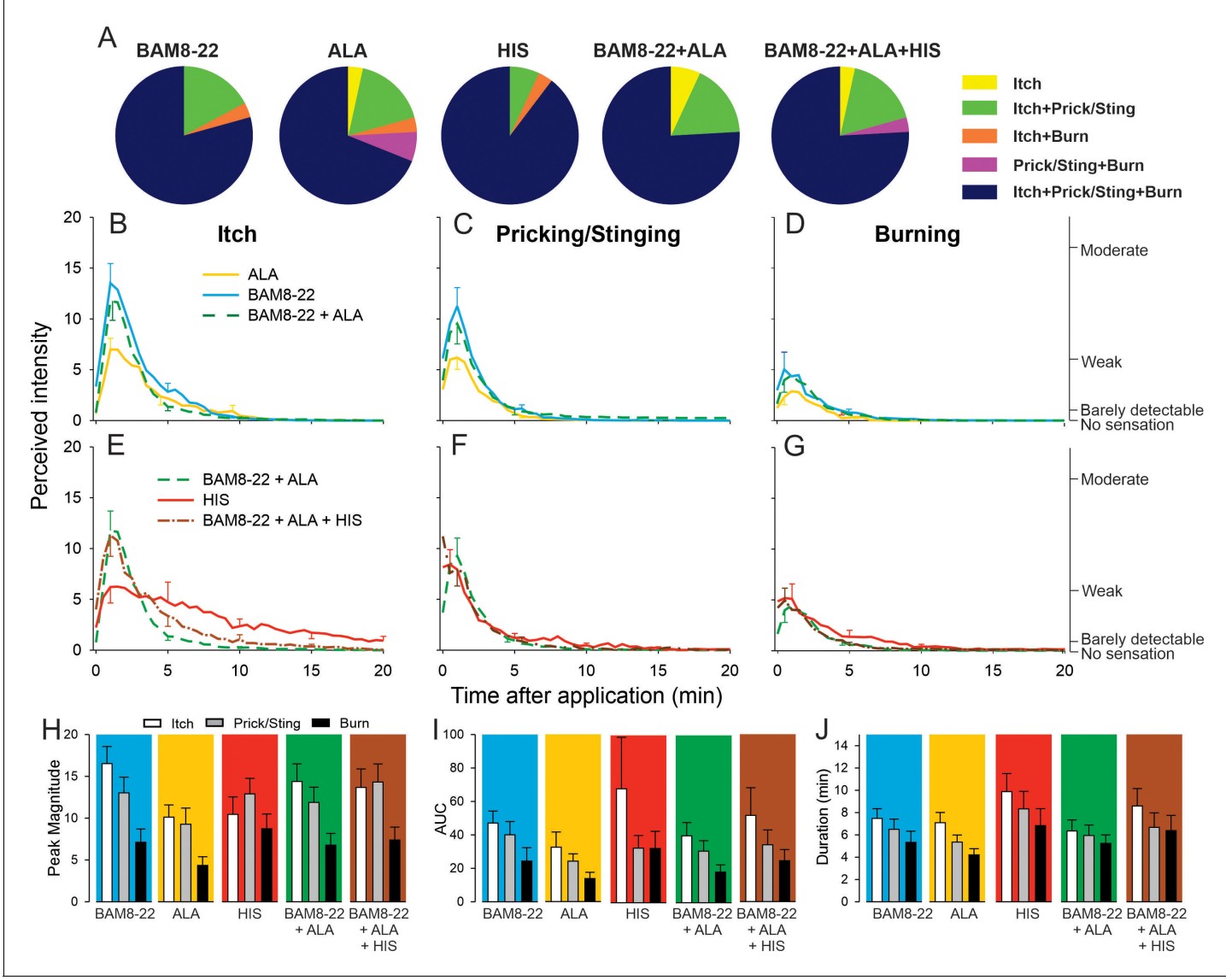

**Figure 5.** In humans, co-injection of β-alanine (ALA) and bovine adrenal medulla peptide 8–22 (BAM8-22) with or without histamine (HIS) does not change the itch or nociceptive sensations compared to the effects of one component pruritogen given alone. (**A**) The majority of subjects (n = 29) reported itch, pricking/stinging, and burning sensations after administration of each of the pruritogens or their combinations. (**B**) Magnitude of itch, (**C**) stinging/pricking, and (**D**) burning sensations evoked by BAM8-22, ALA, and a combination of BAM8-22 and ALA are plotted for successive 30 s intervals after injection averaged across all 29 subjects. (**E**) Magnitude of itch, (**F**) stinging/pricking, and (**G**) burning sensations evoked by HIS, a combination of BAM8-22 and ALA, and a combination of BAM8-22, ALA, and HIS. For clarity, the standard error of the mean (SEM) is plotted only every 5 min starting with the peak rating for each quality. On the right vertical axis, the locations of three verbal descriptors are shown in correspondence with the ratings of perceived intensity indicated on the left vertical axis (see Materials and methods). (**H**) Peak magnitude of perceived sensation intensity (n = 29), (**I**) computed area under the curve (n = 17), or (**J**) duration of itch, stinging/pricking, or burn ratings (n = 17) were each not significantly different for the five different stimuli, that is, three single pruritogens and two combinations (repeated measures ANOVA [RMANOVA], see text for further details). For (**I**) and (**J**), subjects (n = 12) without a measurable area under the curve (AUC) or duration of any sensation during the time course were not included in the analysis, and therefore only 17 subjects were used for within-subjects analysis and post hoc comparisons. Data represent mean ± SEM.

The online version of this article includes the following source data and figure supplement(s) for figure 5:

**Source data 1.** Magnitude of sensation, AUC and Duration of sensation in response to single pruritogen or their combinations for itch, prcking/stinging and burning.

**Figure supplement 1.** In humans, co-injection of β-alanine (ALA) and bovine adrenal medulla peptide 8–22 (BAM8-22) with or without histamine (HIS) does not change the itch or nociceptive sensation compared to the effects of one of the component pruritogens given alone.

**Figure supplement 1—source data 1.** Magnitude of of sensation, AUC and Duration of sensation in response to single puritogen or their combinations for itch and nociceptive sensation.

significance. Therefore, 'sex' was excluded from further analysis, and the RMANOVAs were re-run only with the within-subjects factors using Bonferroni tests for post hoc testing. For peak ratings (n = 29 subjects), within-subjects factors 'sensory quality' and 'stimulus' were significant (respectively, $F_{(1.67,46.67)}$ = 12.13, p<0.001 and $F_{(3.31,92.80)}$ = 3.39, p<0.018). The mean peak ratings for itch and pricking/stinging were each significantly greater than those for burning (p<0.001, Bonferroni test), and the mean peak ratings of itch and pricking/stinging did not differ significantly. For the factor 'stimulus,' mean peak ratings were significantly different only between ALA and BAM8-22 (p=0.01), and between ALA and the combined administration of all three pruritogens (p=0.03). The mean peak ratings for each sensory quality and stimulus are shown in *Figure 5H*, but the interaction of 'stimulus' and 'sensory quality' was not significant ($F_{(4.82,135.02)}$ = 2.11; p=0.07).

Comparisons of mean AUCs and durations between stimuli required subjects to have peak ratings greater than zero in response to each stimulus (because there cannot be a duration of a sensation that was absent, i.e., had zero magnitude). Seventeen subjects had nonzero ratings of all three sensory qualities in response to each stimulus. Only 'sensory quality' was significant for AUC and duration ($F_{(1.33,21.32)}$ = 4.89, p=0.029 and $F_{(1.46,23.39)}$ = 9.18, p=0.003, respectively, RMANOVA). The mean AUC was significantly larger for itch than for burning (p=0.012, Bonferroni test), and itch lasted significantly longer than burning (p<0.001, Bonferroni test). No statistical differences were found between AUCs and durations of itch, pricking/stinging, and burning evoked by BAM8-22, ALA, HIS, and combinations thereof (*Figure 5I, J*).

Alternative analyses of all 25 subjects that had a nonzero rating of both itch and either of two nociceptive qualities whichever had the greater AUC revealed similar results (see *Figure 5—figure supplement 1*). Thus, a combination of pruritogens did not alter the magnitude or quality of sensations over those elicited by one of its component single pruritogens. For example, increasing the number of activated mechanosensitive nociceptive afferents by combining BAM8-22 and ALA and then adding in MIAs with HIS included in the mixture did not diminish itch and increase the magnitude of nociceptive sensations.

To rule out the possibility that any one pruritogen would dominate any of the sensations when a combination was applied, we re-ran the same analysis after selecting for subjects in whom the lowest maximum peak itch rating to any single pruritogen was at least 50% of the highest maximum peak itch rating to any single pruritogen. Analysis of data from subjects that fulfilled this criterion (n = 11) did not lead to a different result than when all subjects were included. Taken together, these results suggest that intradermal injection of combinations of pruritogens does not lead to an increased sensation of itch or, alternatively, to a diminished sensation of itch with an increase in nociceptive sensations.

To test for whether any mean areas of dysesthesia, wheal, and flare differed between stimuli, separate RMANOVAs were performed for each type of area (*Figure 6*). There was a significant main effect of 'stimulus' for alloknesis ($F_{(2.18,61.08)}$ = 10.27, p<0.001), hyperalgesia ($F_{(2.97,83.04)}$ = 12.79, p<0.001), hyperknesis ($F_{(3.10,86.72)}$ = 7.14, p<0.001), wheal ($F_{(2.53,70.75)}$ = 53.42, p<0.001), and flare ($F_{(2.08,58.15)}$ = 32.94, p<0.001). The mean areas of alloknesis and hyperalgesia after HIS and the combined administration of the three pruritogens were significantly greater compared to those induced by BAM8-22, ALA, or BAM8-22+ALA (*Figure 6A*, see *Supplementary file 2* for p-values). For hyperknesis, HIS and the triple combination of pruritogens each induced significantly larger areas than either ALA or BAM8-22 alone, but not compared to the combination of BAM8-22+ALA. Notably, the combination of BAM8-22+ALA did not demonstrate any significant increase in alloknesis, hyperalgesia, or hyperknesis compared to BAM8-22 or ALA administered alone. The combination of three pruritogens also did not significantly enhance any of the dysesthesia areas compared to HIS alone.

The mean areas of wheal and flare were significantly larger in response to HIS and to the combination of all three pruritogens than in response to either BAM8-22 or ALA alone, or their combination (*Figure 6B*; see *Supplementary file 2* for p-values). Furthermore, the combination of BAM8-22 + ALA did not significantly alter the areas of flare and wheal compared to each compound alone. Similarly, the triple combination did not cause larger areas of wheal and flare than HIS alone.

We conclude that the co-activation of different classes of CMHs or co-activation of CMHs and C-MIAs does not increase areas of dysesthesias, flare, and/or wheal in comparison to the activation of distinct classes of nociceptors alone.

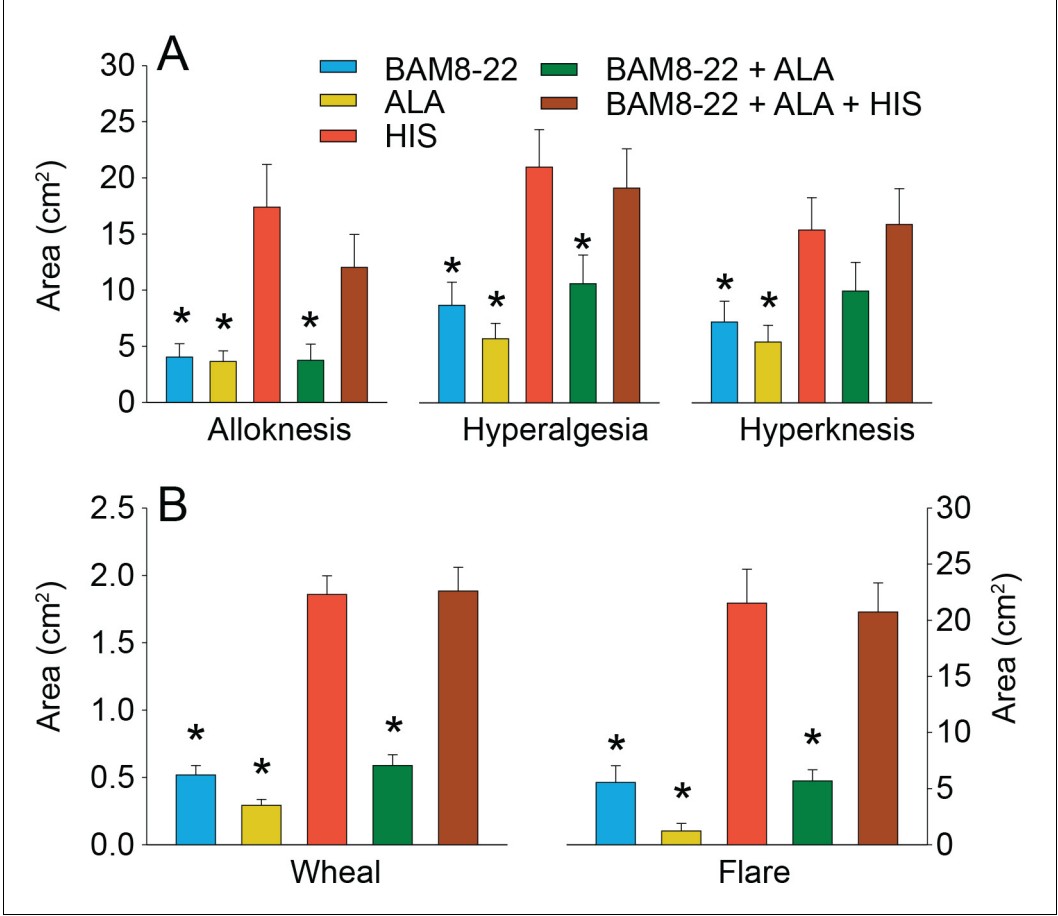

**Figure 6.** In humans, co-injection of β-alanine (ALA) and bovine adrenal medulla peptide 8–22 (BAM8-22) with or without histamine (HIS) does not change the areas of dysesthesia or skin reactions compared to the effects of one of the component pruritogens given alone. (**A**) Areas of alloknesis, hyperalgesia, and hyperknesis and (**B**) areas of wheal and flare produced after injection of each pruritogen (BAM8-22, ALA), HIS, and combination of pruritogens. Data were analyzed with repeated measures ANOVA (RMANOVAs). Greenhouse–Geisser corrections were made for all analyses to correct for non-sphericity followed by Bonferroni post hoc tests (see main text and *Supplementary file 2* for p-values). * indicates significant differences compared to HIS or BAM8-22 + ALA + HIS. Data from all subjects (n = 29) were included in the analysis. Data represent mean ± standard error of the mean (SEM).

The online version of this article includes the following source data for figure 6:

**Source data 1.** Areas of dysesthesias, wheal and flare in response to a single pruritogen or their combination.

## Discussion

In our study, we investigated how itch is coded in primates by examining the expression pattern of pruritogen receptors in human and macaque DRG, by determining physiological responses of non-human primate pruriceptive nociceptors, and by assessing psychophysical effects of pruritogen mixtures on itch perception in human subjects. Specifically, we explored the expression of MRGPRX1 and the physiological effects of its agonist BAM8-22, comparing these with MRGPRD expression and responses of the MRGPRD-activating pruritogen, ALA. Employing ISH, we found that MRGPRX1 and MRGPRD are predominantly co-expressed in human DRG, indicating that primate and mouse pruritogen receptors have different patterns of expression. Using electrophysiology, we discovered that the MRGPRX1 and MRGPRD agonists activated predominantly cutaneous polymodal C-fibers with some differences in BAM8-22 activating almost all and ALA stimulating a smaller defined subset of C-fibers. Lastly, taking advantage of the fact that ALA, BAM8-22, and HIS differentially activate certain types of C-fibers, we investigated sensory coding of itch in humans. We observed that

mixtures of these pruritogens elicit itch percepts similar to those evoked by singly applied pruritogens, suggesting that combined versus differential fiber activation does not have a large effect in shaping the quality of sensation.

Extensive genetic manipulation studies in mice have revealed a great deal about the receptors and sensory neurons that underlie itch in mice (*Hoon, 2015*; *Lay and Dong, 2020*). Although less is known about the neuronal mechanisms of itch in primate, it is often assumed that they are similar to those in rodent. An example of this relates to itch induced by the BAM8-22 peptide, a cleavage product of proenkephalin, shown to be a pruritogen in a human psychophysical study (*Sikand et al., 2011a*) that also triggers scratching in mice. In rodents, BAM8-22 activates a receptor named *Mgprc* (*Grazzini et al., 2004*; *Han et al., 2002*). Suggestive of a labeled line for itch, this receptor is expressed in a select population of DRG neurons that are required for BAM8-22-induced scratching and sufficient to evoke scratching when activated (*Han et al., 2013*). However, primates lack the *Mgprc* receptor or a differently named orthologous receptor and instead harbor a distinct subfamily of MRGPRs, named MRGPRX (*Solinski et al., 2014*), that is, clear orthologous receptors of this subfamily are absent in rodents (*Zylka et al., 2003*). Despite this major difference in pruritogen receptor families between primates and rodents, a striking pharmacological similarity exists between receptors in primates and mice, with MRGPRX1 in humans, like *Mgprc* in mice, being activated by BAM8-22 (*Lembo et al., 2002*) and MRGPRD being activated by ALA (*Liu et al., 2012*; *Shinohara et al., 2004*). Therefore, we sought to examine the expression profile of MRGPRX1 and MRGPRD in human DRG. Similar to the expression of *Mgprc* in mice, we found that MRGPRX1 is expressed in a subpopulation of human TRPV1-positive neurons. However, we found that the MRGPRD expression largely overlaps with MRGPRX1 in human DRG. We did not directly quantify the number of neurons expressing MRGPRD or MRGPRX1 in human DRG neurons. However, in a previous study (*Solinski et al., 2019a*), we found that the number of NPPB+ human DRG neurons (9.8%) corresponds well with the number of *Nppb*+ neurons found in mouse DRG neurons. As about four out of five NPPB+ human DRG neurons co-express MRGPRX1, the MRGPRX1/MRGPRD+ population is about 8% of all DRG neurons. Of note, we also observed the overlapping expression of MRGPRD and MRGPRX1 in macaque DRG neurons, indicating a conserved expression pattern in the primate lineage. However, this expression pattern is dramatically different from that found in mice, where *Mgprd* is present in a distinct subclass of neurons that do not express *Mgprc* and do not express *Trpv1* (*Zylka et al., 2003*). These results point to primate pruriception in peripheral sensory neurons being different from that in mice. Previously, we reported that MRGPRX1-positive human DRG neurons express NPPB and that these cells express additional pruritogen receptors, HRH1 and IL31RA (*Solinski et al., 2019a*). Again, this expression pattern is different from that found in mice where *Nppb* and *Il31ra* are expressed in a class of neurons distinct from those that express either *Mgprc* or *Mgprd* (*Dong et al., 2001*; *Nguyen et al., 2017*; *Usoskin et al., 2015*; *Zylka et al., 2003*; *Zylka et al., 2005*). Interestingly, *Nppb* was shown to be necessary for the transmission of pruritic signals between primary afferents and second order spinal cord neurons in mice (*Mishra and Hoon, 2013*), which in turn is suggestive of the idea that the MRGPRX1/MRGPRD/NPPB neuronal population is also involved in primate itch. Thus, our findings expand our previous knowledge on human sensory neurons and indicate that, at least at the level of the DRG soma, pruritogen receptors are expressed in a largely overlapping fashion in humans.

To complement our studies examining the expression of pruriceptive receptors in human DRG, we stimulated cutaneous afferents in monkeys with BAM8-22 and ALA and measured evoked neuronal activity using a teased-fiber preparation. Previously, we reported that pruritogens activate a number of physiologically distinct subtypes of nerve fibers, including cutaneous CMHs, C-MIAs, as well as A-fiber nociceptors (*Ringkamp and Meyer, 2014*). In addition, we showed that cutaneous CMHs can be distinguished into two types based on their responses to a stepped heat stimulus, namely QC and SC fibers (*Johanek et al., 2008*; *Meyer and Campbell, 1981*; *Wooten et al., 2014*). Interestingly, ALA or BAM8-22 did not activate A-fiber nociceptors, and C-MIAs were activated rarely by BAM8-22 (2/6 responsive), and not at all by ALA. Because BAM8-22 could elicit a small flare (~5 cm$^2$ on average) in our psychophysical experiments, we cannot rule out the possibility of an indirect C-MIA activation via mast cells (*Solinski et al., 2013*; *Subramanian et al., 2011*). However, taken together, these findings suggest that C-MIAs and A-fiber nociceptors are not major contributors to sensations produced by ALA and BAM8-22.

In contrast, we show that CMHs are excited by intradermal injections of ALA and BAM8-22. Confirming our earlier results (*Wooten et al., 2014*), we found that the MRGPRD agonist ALA preferentially activated QCs, indicating that about 60% of cutaneous CMHs (nearly all QCs but only 23% of SCs) are sensitive to ALA in monkeys. In agreement with our human expression data, BAM8-22 also activated a large proportion of QCs (72% of QCs responded to BAM8-22), indicating that MRGPRD and MRGPRX1 are co-expressed in this functional class of cutaneous primary afferents. Surprisingly, given our expression data, we observed that BAM8-22 evoked APs in the SC class of CMHs (93% of SCs responded to BAM8-22) that was largely unresponsive to ALA. Furthermore, BAM8-22-induced responses in SCs were about twice as large as in QCs.

Several explanations could account for this apparent discrepancy between our expression and functional data: (1) species-specific differences, between monkeys and humans, in the DRG expression pattern of MRGPRD/MRGPRX1 might exist. In fact, our comparative analysis of MRGPRD/MRGPRX1 expression in human and primate DRG supports this explanation and points to MRGPRD marking a subpopulation of the bigger MRGPRX1+ group of neurons in macaques. This pattern is consistent with a recent report characterizing the rhesus macaque DRG with single-cell transcriptomics (*Kupari et al., 2021*). (2) While ISH was used to study MRGPR expression at the neuronal soma, our physiological assay probes the functional activation of receptors in cutaneous axonal terminals. It is conceivable that mRNA expression in the soma might not be a relevant predictor for functional receptor in the periphery. (3) Potency and efficacy of GPCR agonists not only depend on the expression of the cognate receptor protein but can be modulated allosterically by binding partners of the receptor and associated proteins involved in AP firing, which could potentially be expressed differentially between QCs and SCs. Thus, we might have missed co-expression of MRGPRD and MRGPRX1 in functional studies due to cell-autonomous pharmacological differences.

That a large proportion (54/66) of CMHs is responsive to BAM8-22 is consistent with previous findings showing that spicules of cowhage, a tropical legume, activated all tested CMHs in human (*Namer et al., 2008*), nearly all QC and SC mechanoheat nociceptors in monkey (*Johanek et al., 2008*), and that mucunain, the pruritic cysteine protease contained in these spicules, activates MRGPRX1 in heterologous cells (*Reddy et al., 2018*). However, although about half of A-fiber nociceptors in nonhuman primate is activated by cowhage spicules (*Ringkamp et al., 2011*), BAM8-22 did not activate this fiber type. This suggests that cowhage responses in these fibers may be mediated through a different mechanism, for example, protease-activated receptors 2 and 4, which mucunain has previously been shown to activate (*Reddy et al., 2008*).

In nonhuman primates, we found that all mechanosensitive unmyelinated cutaneous afferents responsive to MRGPR agonists were also activated by heat, and we found previously that QCs and SCs are sensitized to heat stimuli following a mild burn injury (*Wooten et al., 2014*). These observations are consistent with the expression of TRPV1 in these neurons and are in agreement with the overlapping expression of TRPV1 and MRGRX1 and MRGPRD that we observed in human DRG. This is in contrast to mice in which about half of the GFP-labeled *Mrgprd*+ neurons in a *Mrgprd* reporter line were unresponsive to heat stimuli (*Liu et al., 2012*; *Qu et al., 2014*). Together, these findings show that there are key differences in the gene expression of pruriceptive receptors between mice, human, and macaques, and in the functional properties of cutaneous unmyelinated afferents between mice and nonhuman primate. Although DRG-enriched genes are conserved between mice and human (*Ray et al., 2018*), our findings extend results from a recent study (*Shiers et al., 2020*) reporting substantial differences in the co-expression of several genes (including, e.g., $P_2X_3$ and CGRP; TRPV1 and TRPA1) between mice and human DRG neurons. These results will in the future likely be supplemented by the advent of single-cell sequencing technologies to determine the transcriptome of DRG neurons in humans. Such studies should also help explain the potential difference in pruriceptive sensitivity between different physiologically defined subclasses of primary afferent neurons.

In electrophysiological experiments, the peak discharge recorded from nociceptive primary afferents in our present and previous (*Wooten et al., 2014*) study and from pruriceptive STT neurons in the monkey (*Davidson et al., 2012*; *Davidson et al., 2007*) occurred within the first 10–30 s after cutaneous application of different pruritogens. In contrast, in psychophysical experiments, the time courses of itch and nociceptive sensations were similar for the different pruritogens with a peak intensity occurring between 1 and 2 min after injection. One possible explanation for this discrepancy is that the delayed peak in sensory intensity is due to the recruitment of additional nerve fibers,

STT- and cortical neurons when the injected pruritogen spreads over time, thereby activating additional primary afferents or central neurons. The peak intensity of sensation would then occur when all primary afferents or central neurons within the reach of the pruritogen are activated.

Primate primary sensory afferents responsive to pruritogens are also activated by mechanical, thermal, and chemical nociceptive stimuli, and activity in these afferents is thought to elicit the sensation of pain from such stimuli (*Ringkamp et al., 2013*). This would rule out the possibility that these primary sensory neurons are specifically activated only by pruritic stimuli. Furthermore, a labeled line for itch has not been identified in human and nonhuman primate. Consequently, several different models explaining how pruritic stimuli may be encoded have been proposed (for review, see *Carstens et al., 2020*; *LaMotte et al., 2014*). Among these, the itch model of 'spatial contrast' posits that a pruritic stimulus is signaled by activity in a few nociceptive afferents among other non-active fibers innervating a given skin region (i.e., spatially sparse), whereas nociceptive stimuli are encoded in the activity of many afferents (*Namer et al., 2008*; *Namer and Reeh, 2013*; *Schmelz, 2010*; *Steinhoff et al., 2019*). This model predicts that the sensation of itch will decrease and the sensation of pain will increase when a larger number of nociceptive afferents is activated from a given site. Our finding of a preferential activation of distinct types of C-fiber nociceptors by ALA, BAM8-22, and HIS in monkey and reports that HIS-induced itch occurs through activation of C-MIAs in humans (*Schmelz et al., 1997*; *Wooten et al., 2014*), permitting us to test a key prediction of the spatial contrast model using these itch-inducing agents in psychophysical experiments. In these experiments, we used BAM8-22, ALA, and HIS, each of which, when injected singly into the skin of subjects, induced itch and to a lower extent pain-like sensations of pricking/stinging and burning. Spatial contrast was then varied by co-injecting combinations of the three pruritogens, each activating different classes of primary afferents in a given skin site. We found that neither the magnitudes of itch nor nociceptive sensations changed between application of a single pruritogen and mixtures, a finding that is incongruous with the 'spatial contrast' model. This finding is also consistent with prior evidence for a lack of spatial summation or enhancement of itch when multiple cowhage spicules applied to a local area of skin did not evoke more itch than a single spicule despite presumably activating more cowhage-sensitive CMHs (*LaMotte et al., 2009*). Similarly, we now found that increasing the number of activated nerve fibers of one type (CMHs with injection of ALA + BAM8-22) by adding those of another type (MIAs with HIS) also did not enhance the magnitude of itch.

Since the 'spatial contrast' model does not explain the results from our psychophysical experiment, what other mechanisms might be responsible for the coding of pruritic stimuli? One previously suggested solution that might explain these results is that itch- and pain-related information is separately encoded at the level of the spinal cord where pruriceptive and nociceptive STT neurons have been identified (*Davidson et al., 2007*; *LaMotte et al., 2009*). Input from nociceptive or pruriceptive afferents may be routed to the pruriceptive and nociceptive STT pathways through dedicated spinal interneuronal circuits. This possibility is supported by the observation that itch-like scratching behavior can be selectively blocked without reducing nociceptive behavior by *Nppb* deficiency (*Mishra and Hoon, 2013*), by the application of antagonists of gastrin-releasing peptide receptor (GRPR) (*Sun and Chen, 2007*), or by eliminating GRPR-neurons (*Sun et al., 2017*; *Sun et al., 2009*). However, pruriceptive STT neurons are also responsive to noxious heat, capsaicin, and mechanical stimuli (*Davidson et al., 2007*). Thus, there must still exist a mechanism by which nociceptive afferents are able to differentially activate nociceptive and pruriceptive STT pathways. Differential activation may be achieved by the level of activity and/or the pattern of discharge that is induced by nociceptive and pruritic stimuli in nociceptive afferents. In support of this view, we have previously shown that capsaicin produced a significantly higher median discharge frequency in C-MIAs than HIS, whereas the total number of induced APs did not differ between the two stimuli (*Wooten et al., 2014*). Similarly, capsaicin produced a higher intraburst frequency discharge in HIS-sensitive STT neurons than HIS (*Davidson et al., 2012*). In addition, activating a combination of different afferent fibers having varying tuning capabilities for particular chemical, mechanical, or thermal stimuli (e.g., QCs and SCs) and different firing patterns together may play a role in the transmission of information needed to discriminate pain from itch. Thus, while we were unable to absolutely identify the mechanism by which itch is distinguished from pain in primates, our studies suggest that the 'spatial contrast' model does not account for this process under normal conditions.

The results of this and previous studies indicate that in normal skin different classes of primary afferents are tuned to preferentially respond to different pruritogens. It is currently unclear whether in chronic itch primary afferents become responsive to a broad range of pruritogens or whether a preferential but increased responsiveness is maintained. Future studies may investigate this question and whether peripherally applied MRGPR-antagonists can provide relief from chronic itch in human. Considering that a large majority of nociceptive afferents responded to ALA and BAM8-22, which produced itch and nociceptive sensations in human, anti-pruritic effects of such compounds appear likely. It will also be interesting to investigate whether MRGPR-antagonists have peripheral anti-nociceptive effects.

In conclusion, our study defines the expression of MRGPRX1 and MRGPRD in human DRG neurons and the electrophysiological responses of their agonists in primate cutaneous primary afferents. In human psychophysical studies, we observed that co-administration of different MRGPR agonists and HIS did not change overall reports of itch and nociceptive sensations. Taken together, the neuronal processing of itch differs between rodent and primate, and additional studies are needed to determine the coding mechanisms of itch and pain in human.

# Materials and methods

**Key resources table**

| Reagent type (species) or resource | Designation | Source or reference | Identifiers | Additional information |
|---|---|---|---|---|
| Strain, strain background (*Macaca nemestrina*) | *Macaca nemestrina* | Johns Hopkins, NIH-supported pigtailed macaque breeding colony | | |
| Biological sample (*Homo sapiens*) | DRG tissue | Tissue For Research (http://www.biobankonline.com) and the NIH NeuroBioBank | | See *Supplementary file 4* |
| Biological sample (pigtailed monkey) | DRG Tissue (*Macaca nemestrina*) | Johns Hopkins, NIH-supported pigtailed macaque breeding colony | | |
| Peptide, recombinant protein | Bovine adrenal medulla peptide 8–22 (BAM8-22) | Tocris | Cat no. 1763 | |
| Peptide, recombinant protein | Bovine adrenal medulla peptide 8–18 (BAM8-18) | Keck Lab, Yale University, New Haven **Sikand et al., 2011b** | | |
| Commercial assay or kit | hTRPV1 in situ hybridization probe | Advanced Cell Diagnostics | Cat no. 415381 | |
| Commercial assay or kit | hMRGPRX1 in situ hybridization probe | Advanced Cell Diagnostics | Cat no. 517011 | |
| Commercial assay or kit | hMRGPRD in situ hybridization probe | Advanced Cell Diagnostics | Cat no. 524871 | |
| Commercial assay or kit | RNAscope multiplex fluorescent development kit | Advanced Cell Diagnostics | Cat no. 320850 | |
| Commercial assay or kit | BaseScope Probe - BA-Hs-MRGPRX1-1zz-st | Advanced Cell Diagnostics | Cat no. 717091 | |

*Continued on next page*

*Continued*

| Reagent type (species) or resource | Designation | Source or reference | Identifiers | Additional information |
|---|---|---|---|---|
| Commercial assay or kit | BaseScope Probe BA-Hs-MRGPRD-1zz-st | Advanced Cell Diagnostics | Cat no. 878101-C2 | |
| Commercial assay or kit | BaseScope Duplex Reagent Kit | Advanced Cell Diagnostics | Cat no. 323800 | |
| Chemical compound, drug | β-Alanine (ALA) | Sigma-Aldrich | 146064-25G, A9920-100G | |
| Chemical compound, drug | Histamine dihydrochloride (HIS) | Sigma-Aldrich | H7250-5G | |
| Chemical compound, drug | Ketamine hydrochloride | Phoenix Pharmaceutical, St. Louis, MO | | |
| Chemical compound, drug | Sodium pentobarbital | Akorn Pharmaceutical, Lake Forest, IL | | |
| Chemical compound, drug | Cefazolin | West-Ward Pharmaceutical Corp, Eatontown, NJ | | |
| Chemical compound, drug | Buprenorphine SR | ZooPharm, Laramie, WY | | |
| Software, algorithm | Data acquisition processing system (DAPSYS), version 8 | Brian Turnquist (Bethel University) *Sikand et al., 2011a*; *Wooten et al., 2014* | | http://www.dapsys.net/ |
| Software, algorithm | SigmaPlot 10 | Systat Software Inc, San Jose, CA | | |
| Software, algorithm | Statistica, version 13 | Tibco | | |
| Software, algorithm | Corel Draw (version 9.439 or 2020) | Corel Corporation, Ottawa, Canada | | |
| Software, algorithm | Adobe Illustrator | Adobe, San Jose, CA | | |
| Software, algorithm | ImageJ | NIH, Bethesda, MD | | |
| Software, algorithm | T-Coffee | EMBL-EBI, Hinxton, UK | | https://www.ebi.ac.uk/Tools/msa/tcoffee/ |
| Software, algorithm | FastTree 2.1 | *Price et al., 2010* | | https://usegalaxy.eu/ |
| Software, algorithm | Newick Display | *Dress et al., 2008* | | https://usegalaxy.eu/ |

## Phylogenetic tree construction

Protein sequences (see *Supplementary file 3*) were obtained from Ensembl and aligned using T-Coffee (https://www.ebi.ac.uk/Tools/msa/tcoffee/) with default settings. From the aligned sequences, the phylogenetic tree was calculated with FastTree 2.1 (*Price et al., 2010*), using the Jones–Taylor–Thornton 1992 and CAT protein evolution model, and visualized with Newick display (*Dress et al., 2008*). FastTree 2.1 and Newick display were invoked from the Galaxy web platform (*Afgan et al., 2018*) using the public server usegalaxy.eu.

## In situ hybridization of human DRG

ISH of human DRG sections was performed as described previously (*Solinski et al., 2019a*). Briefly, postmortem DRGs from four different, healthy donors (see *Supplementary file 4*) were obtained from Tissue For Research (https://biobankonline.com/) and the NIH NeuroBioBank at the University of Maryland (Baltimore, MD). ISH staining was performed on 14–20 μm cryo-sections with probes specific for *hTRPV1 (#415381)*, *hMRGPRX1 (#517011)*, and *hMRGPRD (#524871)* in conjunction with the RNAscope multiplex fluorescent development kit (ACD) according to the manufacturer's instructions. Nuclei were counter-stained using DAPI. Images were collected on an Eclipse Ti (Nikon) confocal laser scanning microscope using a ×40 objective.

## In situ hybridization of macaque DRG

Three DRGs were collected postmortem from one pigtail monkey, flash frozen in liquid nitrogen, and stored at −80˚C. ISH was performed on 14 μm cryo-sections with custom probes specific for MRGPRX1 and MRGPRD in conjunction with the BaseScope duplex assay (Advanced Cell Diagnostics, Inc, Newark, CA) according to the manufacturer's instructions. Nuclei were stained using hematoxylin. Images were collected on a Keyence microscope (Model BZ-X) using a ×40 objective.

## Primate electrophysiology: surgical preparation

Male and female pigtail monkeys (*M. nemestrina*, n = 8 males and 4 females) were sedated by ketamine hydrochloride (Ketaject, St. Joseph, MO; 12 mg/kg with 0.04 mg/kg atropine) followed by bolus and a continuous venous infusion of sodium pentobarbital to maintain anesthesia (Nembutal, Akorn Pharmaceuticals, Lake Forest, IL; ~5 mg/kg/hr). A saline drip containing 5% d-glucose was continually infused to maintain hydration, and Cefazolin (West-Ward Pharmaceutical Corp, Eatontown, NJ; 20 mg/kg) was administered every 2 hr for antibiotic prophylaxis. Body temperature was maintained by feedback-controlled heating pads (~38˚C), and heart rate was monitored by an electrocardiogram. If heart rate increased more than 10% upon noxious stimulation, a bolus of pentobarbital was administered. Animals were intubated, paralyzed with pancuronium bromide (Sigma-Aldrich, St. Louis, MO) and ventilated at a rate to maintain an expiratory $pCO_2$ near 40 mmHg. At the end of the experiments, animals received buprenorphine SR (0.2 mg/kg sc, ZooPharm, Laramie, WY) as prophylactic analgesic treatment before being returned to their home cage.

Radial, medial cutaneous, sural, peroneal, and saphenous nerves were used for teased-fiber recordings as described previously (*Campbell and Meyer, 1983*). Briefly, a given nerve was aseptically dissected under a microscope, and skin edges of the incision were sutured to a metal ring to form a pool that was filled with paraffin oil. To record neuronal activity from single nociceptive afferents, small nerve bundles were cut from the nerve proximally, teased into smaller filaments on a dissecting platform, and wrapped around a silver-wire recording electrode placed above the splitting platform. Teasing of a filament continued until neuronal activity could be recorded from single nociceptive A- and C fibers. A stimulating electrode located ~4 cm distally at the nerve trunk was used to electrically stimulate the nerve to reveal fibers on the recording electrode and to determine the conduction velocity at the nerve trunk of the afferent under study. RFs were located by applying gentle squeeze stimuli to the skin innervated by the nerve under study. RFs of nerve fibers unresponsive to such stimuli were located by electrical stimuli applied transcutaneously via a pair of ball electrodes using a search paradigm described previously (*Meyer et al., 1991*; *Ringkamp et al., 2001*). In fibers responsive to gentle squeezing, the mechanosensitive RF was mapped with a 8.9 bar von Frey hair (Stoelting, Wood Dale, IL, USA), and boundaries were marked on the skin with a felt tip permanent marker. Using a slightly suprathreshold von Frey hair, the most sensitive, mechanoresponsive spot was identified within the RF and marked as the hot spot. von Frey hairs of increasing stiffness were applied to this spot to determine the mechanical threshold of the nerve fiber. The smallest von Frey hair producing a neuronal response in two out of four applications was regarded as threshold. Units were classified as mechano-sensitive afferents (MSAs) if they were responsive to von Frey hair stimuli <6 bar, and mechano-insensitive afferents (MIAs) if the threshold was >6 bar. Transcutaneous electrical stimulation at the proximal edge of the RF was used to measure conduction latencies from the skin. Stimuli up to 100 mA were used, and the latency at this intensity was regarded as the conduction latency of the peripheral parent axon. Conduction distance was measured from the RF to the recording electrode using suture material placed along the presumed path

of the nerve, and conduction velocities were calculated by dividing this distance by the conduction latency recorded from electrical stimulation. Fibers with a conduction velocity <2 m/s were classified as C- and those between 2 and 20 m/s as Aδ-fibers. The responsiveness to heat was tested by applying a heat stimulus (49°C, 3 s from a base temperature of 38°C for 3 s) to the middle of the RF using a feedback-controlled carbon dioxide laser (*Meyer et al., 1976*). Instantaneous frequencies of responses to this stimulus were three point median smoothed and then used to classify C fibers as quickly (QC) or slowly (SC) responding afferents as previously described (*Wooten et al., 2014*). Briefly, QCs responded with a burst of discharge at the onset of the heat stimulus, and the response then quickly adapted during the temperature plateau. In contrast, SCs responded slowly, lacked a burst discharge at the response onset, and the discharge peaked during the plateau phase of the heat stimulus. Ten minutes after this stepped heat stimulus, a heat staircase was applied (baseline of 38°C for 15 s, followed by 1°C, 1 s increments up to 49°C) to measure heat threshold.

## Primate electrophysiology: chemical stimulation

Ten minutes after applying the heat staircase, two blocks of intradermal injections (each 10 µl) were administered at the RF. One block consisted of extracellular fluid (ECF, the vehicle) followed by ALA (90 µg in ECF, Sigma-Aldrich, St Louis, MO), and another block of BAM8-18, the control peptide (1 µg in saline, synthesized in the Keck Lab, Yale University, New Haven, CT), followed by BAM8-22 (1 µg in saline, Tocris Bioscience). Each injection (volume 10 µl) was administered with a 28 G1/2 Lo-Dose syringe (Becton Dickenson, Franklin Lakes, NJ). The order of the blocks was randomized. Following each injection, neuronal activity was recorded for at least 5 min or until no neuronal activity was recorded for three consecutive minutes.

## Primate electrophysiology: data recording

APs were filtered, amplified, and digitized and stored for offline data analysis using a Digital Acquisition Processor board (Microstar Laboratories Inc Bellevue, WA) and DAPSYS software (Brian Turnquist, Bethel University, http://www.dapsys.net/, version 8.78). DAPSYS software was also used to control the $CO_2$ laser stimulator, record the skin temperature, trigger electrical stimuli, and time intradermal injections. APs, trigger pulses, commands for the laser, and injection profiles were 'time-stamped' in DAPSYS and therefore allowed to relate neuronal events and stimuli applied to the RF.

## Human psychophysics: subjects and pruritogens

32 healthy volunteers, 15 females and 17 males, were enrolled after providing written informed consent to protocols approved by the Yale University Human Investigation Committee and that met the Declaration of Helsinki Principles for human subjects. The mean age was 32.8 + 14.9, the median 28 years. Among these 19 were Caucasian, 8 Asian, 2 Hispanic, and 3 were more than one race. Initially, 33 subjects were recruited, 1 did not qualify for participation due to not feeling itch during tests with a single pruritogen and data from 3 subjects were not included in the analysis because all stimuli could not be tested due to scheduling issues.

Potential subjects were excluded from participation in the study if they reported a medical history of cardiac, immunological, dermatological, or neurological disorders or reported taking antihistamine or analgesic medication recently.

The pruritogens, ALA (Sigma-Aldrich; 90 µg/10 µl; 100 mM; 1.01 µmol), BAM8-22 (Tocris Bioscience; 1 µg/10 µl; 51 µM; 0.51 nmol), and HIS dihydrochloride (Sigma-Aldrich; 10 µg/10 µl; 5.4 mM; 54 nmol), as well as a combination of two substances (ALA 90 µg with BAM8-22 1 µg in 10 µl) and combination of all three substances (ALA 90 µg with BAM8-22 1 µg and HIS 10 µg in 10 µl) were each prepared under sterile conditions in saline 0.9%. Effects of vehicle solutions (saline, ECF) were not tested as in a previous study normal saline elicited a transient itch of low magnitude in a small proportion of subjects tested (*Simone et al., 1987*). Similarly, in a previous study (*Sikand et al., 2011b*) ECF-induced sensations were minimal with an average peak magnitude itch that did not differ from itch induced by $10^{-3}$ µg of HIS and nociceptive sensations less than those produced by HIS doses $\geq 10^{-2}$ µg. In the present experiments, we did not measure the pH of each solution. But in a previous study no difference was found in the itch evoked by HIS in a vehicle of normal saline alone (with pH of 5.0) versus a vehicle of phosphate buffered saline that had a pH of 7.4 (*Simone et al., 1987*). The testing procedure was performed as previously described (*LaMotte et al., 2009*;

*Sikand et al., 2009*), and concentrations for injections were based on previous studies (*Liu et al., 2012*; *Sikand et al., 2011a*; *Simone et al., 1991*). After disinfection with an ethanol wipe, subjects received a superficial intradermal injection of 10 µl pruritogen solution into the volar forearm with the needle inserted approximately parallel to the skin and the tip of the needle visible through the skin (*Sikand et al., 2011b*). Each arm was tested maximally up to two times per session with subsequent injections made well outside the area of dysesthesias produced by a prior injection and after the sensation of the previous injection was not present anymore. The order of administration for trials was random and blinded for each subject. Control peptides, amino acids, and vehicles (BAM8-18, L-alanine, and saline) were not administered as their evoked responses were either minimal or absent in previous human psychophysics studies (*Liu et al., 2012*; *Sikand et al., 2011a*; *Simone et al., 1987*).

Subjects were instructed not to scratch or move the arm following the injection. A visual barrier was placed following the injection to cover the view on the injected arm, removing biases in ratings of sensations by observed visual changes to the skin.

## Human psychophysics: ratings of sensations after pruritogen injection

Standardized oral instructions explaining the rating, sensations to be rated, and the testing procedure were read to the subjects. Prior to testing, in an independent training session, subjects were familiarized with and instructed to rate the magnitude of sensory quality by moving a cursor to a desired position along the Labeled Magnitude Scale (gLMS) (*Green et al., 1996*) displayed on a computer screen. The display and recording of ratings were controlled by DAPSYS 6.0 (http://www.dapsys.net; Brian Turnquist, Bethel University, St. Paul, Minnesota). The computer converted the chosen position to a numerical score (not visible to the subject) between '0' at the bottom of the line to '100' at the top. The scale was labeled with a quasi-logarithmical distribution of descriptive labels: 'no sensation' (located at a numerical position of 0), 'barely detectable' (1), 'weak' (6), 'moderate' (17), 'strong' (35), 'very strong' (53), and 'strongest imaginable sensation of any kind' (100). Each rating value on the gLMS corresponded with the maximum sensation felt by the individual within the last 30 s interval for each of three distinct sensation categories: 'itching,' 'pricking/stinging,' and 'burning' (*LaMotte et al., 2009*). Itch was defined as a sensation that evokes a desire to scratch. Pricking/stinging was defined as a nociceptive sensation that was sharp and well-localized, either intermittent like a needle prick, or continuous like an insect sting. Burning was defined as a nociceptive sensation most often associated with sunburns or thermal burns, but also with skin abrasions, strong cold, and/or chemical irritants. Instructions also distinguished the above sensations from others, and subjects were specifically asked to rate the magnitude of nociceptive ('pain-like') sensations regardless of whether or not overt pain occurred. Subjects were asked to ignore the initial prick of the needle entering the skin immediately after the injection. Subjects were prompted to rate levels of sensations experienced only in and surrounding the area of the application of the pruritogen every 30 s for a minimum of 5 and a maximum of 20 min or until each sensory quality received three successive ratings of zero magnitude.

## Human psychophysics: skin reactions and areas of dysesthesias

After the conclusion of the sensory ratings, the visual barrier was removed and superficial changes to the skin's appearance were marked on the skin. These could include a wheal (raised edematous region) and/or a 'flare' herein broadly defined to include visible erythema that appears confined to the immediate vicinity of the injection site or extends to a larger area surrounding the site.

Next, subjects were tested for the presence of mechanically evoked dysesthesias surrounding the injection site. Mechanical testing proceeded in a radial manner beginning on the outer edge of the arm and moving inward nearing the center of the injection. Alloknesis was denoted as an area within which itch was evoked by light stroking with a cotton swab. Hyperalgesia and hyperknesis were defined as enhanced pricking pain and itch evoked by von Frey-type filaments with tip diameters (and exerting bending forces) of 200 µm (80 mN) and 50 µm (20 mN), respectively, each applied for 1 s. The borders of each area were marked with a felt tip pen.

The marked areas of changes in skin appearance and dysesthesia were photographed with a scale for further offline analysis and measurement using ImageJ (version 1.51, NIH, Bethesda, MD, USA).

## Data analysis

### In situ hybridization on human tissue

For each double-labeling experiment, sections from four different tissue donors were stained. To avoid bias, complete DRG sections were imaged, and individual images stitched together using NIS elements (Nikon) to generate an image of the whole DRG. Initially, fluorescent channels of the stitched images were handled separately and visually analyzed for neuronal expression of stained markers in ImageJ. A neuron, defined by a broad DAPI-positive neuronal nucleus and/or a dense array of surrounding DAPI-positive satellite glia nuclei, was manually counted as positive only when >5 puncta per cell were present. Post counting individual fluorescent channels, the channels were merged, and potential co-expression of marker genes was assessed permitting blind quantification of co-expression. Data are expressed as mean percentage and standard error of the mean calculated across the four tissue donors and as aggregated number of positive neurons from all four tissue donors.

### In situ hybridization on macaque tissue

Expression analysis for both markers was performed in 1–5 sections from three DRGs, dissected from one macaque. Complete DRG sections were analyzed with ImageJ. A neuron, identified by a dense array of surrounding hematoxylin-stained satellite glia nuclei, was manually counted as positive only when >3 puncta per cell were present. Data are expressed as mean percentage and standard error of the mean calculated across the three DRGs and as aggregated number of positive neurons from all three DRGs.

### Primate electrophysiology

For each injection, spontaneous activity during the 1 min of baseline recording prior to the injection was prorated for the duration of the recorded response and subtracted from the total number of APs recorded following the injection. Responses from the control compound were subtracted to calculate a net response to the pruritogen. A unit was classified as being responsive to a pruritogen if the net response was $\geq 10$ APs/5 min. Data were analyzed with parametric (ANOVA) and non-parametric ($\chi^2$) tests where appropriate (see text for details).

### Human psychophysics

For each subject's ratings of each sensory quality, the following parameters were obtained: (1) the peak rating (highest rating of magnitude), (2) the duration of the sensation (elapsed time between the onset of nonzero ratings and the first rating of zero rating after the sensation disappeared), and (3) the area under the rating curve over time (AUC) for the duration of the sensation. For each of the three measurements and each area of any dysesthesia, flare or wheal, statistical comparisons of means in response to each single pruritogen and the two combinations for pruritogens were analyzed using within-subject RMANOVAs (see main text and Supplementary data for details). Since these data failed the test for sphericity, the Greenhouse–Geisser correction was applied. The Bonferroni test was used for post hoc testing. All electrophysiological and psychophysical data were analyzed with Statistica 13 (TIBCO Software Inc, Palo Alto, CA). A value of $p<0.05$ was considered as statistically significant. Figures were prepared with SigmaPlot 10 (Systat Software Inc, San Jose, CA) and, when necessary, modified with CorelDraw (version 9.439 or 2020, Corel Corporation, Ottawa, Canada) and produced in their final version with Adobe Illustrator. Data are presented as mean ± SEM.

## Acknowledgements

This study was supported by NIH grants R01 AR070875 (MR, RHL) and K01 DA042902 (AHK) and 2T32NS070201 (EIS) and U42OD013117 (Johns Hopkins, NIH-supported pigtailed macaque breeding colony); by the German Research Foundation (DFG): project grant FOR 2690 (HJS) and Research Fellowship grant 326726541 (NMM), the intramural research program of the National Institute of Dental and Craniofacial Research, National Institutes of Health, project ZIADE000721-18 (MAH), and the Neurosurgery Pain Research Institute at the Johns Hopkins University. Human tissue was obtained from the NIH NeuroBioBank at the University of Maryland, Baltimore, MD.

## Additional information

### Competing interests

Xinzhong Dong: Xinzhong Dong is a co-founder and scientific advisor of Escient Pharmaceuticals which is developing drugs targeting Mrgprs. The other authors declare that no competing interests exist.

### Funding

| Funder | Grant reference number | Author |
| --- | --- | --- |
| National Institute of Arthritis and Musculoskeletal and Skin Diseases | R01070875 | Robert H LaMotte Matthias Ringkamp |
| National Institute on Drug Abuse | K01 DA042902 | Amanda Klein |
| Deutsche Forschungsgemeinschaft | FOR 2690 | Hans Jürgen Solinski |
| Deutsche Forschungsgemeinschaft | Fellowship grant 326726541 | Nathalie Malewicz |
| National Institute of Dental and Craniofacial Research | ZIADE000721-18 | Mark A Hoon |
| National Institute of Neurological Disorders and Stroke | T32NS070201 | Elizabeth I Sypek |

The funders had no role in study design, data collection and interpretation, or the decision to submit the work for publication.

### Author contributions

Amanda Klein, Hans Jürgen Solinski, Data curation, Formal analysis, Investigation, Writing - original draft, Writing - review and editing; Nathalie M Malewicz, Hada Fong-ha Ieong, Elizabeth I Sypek, Steven G Shimada, Timothy V Hartke, Matthew Wooten, Data curation, Formal analysis, Investigation, Writing - review and editing; Gang Wu, Data curation, Investigation, Writing - review and editing; Xinzhong Dong, Supervision, Funding acquisition, Investigation, Writing - review and editing; Mark A Hoon, Conceptualization, Data curation, Formal analysis, Supervision, Investigation, Writing - original draft, Writing - review and editing; Robert H LaMotte, Matthias Ringkamp, Conceptualization, Data curation, Formal analysis, Supervision, Funding acquisition, Investigation, Writing - original draft, Project administration, Writing - review and editing

### Author ORCIDs

Amanda Klein (ID) https://orcid.org/0000-0002-3433-2180
Hans Jürgen Solinski (ID) https://orcid.org/0000-0001-6606-3731
Nathalie M Malewicz (ID) https://orcid.org/0000-0002-9045-203X
Hada Fong-ha Ieong (ID) https://orcid.org/0000-0002-3550-8408
Elizabeth I Sypek (ID) https://orcid.org/0000-0001-6904-2426
Steven G Shimada (ID) https://orcid.org/0000-0001-6357-0017
Timothy V Hartke (ID) https://orcid.org/0000-0002-1329-3418
Matthew Wooten (ID) https://orcid.org/0000-0001-8629-9899
Gang Wu (ID) https://orcid.org/0000-0002-6540-4407
Xinzhong Dong (ID) https://orcid.org/0000-0002-9750-7718
Mark A Hoon (ID) https://orcid.org/0000-0002-8794-1684
Robert H LaMotte (ID) https://orcid.org/0000-0002-9079-8639
Matthias Ringkamp (ID) https://orcid.org/0000-0001-6327-0225

## Ethics

Human subjects: Human subjects were enrolled after providing written informed consent to protocols approved by the Yale University Human Investigation Committee and that met the Declaration of Helsinki Principles for human subjects.

Animal experimentation: Animal experiments were approved by the Johns Hopkins University Animal Care and Use Committee (protocol # PR18M295). Experiments were performed in accordance with Office of Laboratory Animal Welfare regulations and the United States Public Health Service Policy on Humane Care and Use of Laboratory Animals.

## Decision letter and Author response

Decision letter https://doi.org/10.7554/eLife.64506.sa1
Author response https://doi.org/10.7554/eLife.64506.sa2

# Additional files

## Supplementary files

• Supplementary file 1. Expression of MRGPRD and MRGPRX1 was assessed in three macaque dorsal root ganglions (DRGs) using double-labeling in situ hybridization (ISH). The number of single- and double-positive neurons is given as aggregated number for each DRG.

• Supplementary file 2. Statistical analysis of areas ($cm^2$) of alloknesis, hyperalgesia, hyperknesis, wheal and flare, or local erythema evoked by injection of a bovine adrenal medulla peptide 8–22 (BAM8-22), β-alanine (ALA), histamine (HIS), a combination of BAM8-22 and ALA, and a combination of BAM8-22 and ALA and HIS. *Bonferroni post hoc p-values compared to HIS. **Bonferroni post hoc p-values compared to BAM8-22 + ALA + HIS. Units are in $cm^2$ and calculated as mean ± standard error of the mean (SEM). Data from all 29 subjects were included in the analysis.

• Supplementary file 3. Protein sequences used for phylogenetic tree construction were fetched from Ensembl. EnsemblGeneIDs, gene names, and the species of the gene are shown.

• Supplementary file 4. Clinical data for human dorsal root ganglion (DRG) donors. DRG tissue was obtained from Tissue For Research (donors 1 and 2) and the NIH NeuroBioBank (donors 3 and 4). Basic clinical data for each donor are summarized.

• Transparent reporting form

## Data availability

All data generated or analysed during this study are included in the manuscript and supporting files. Source data files have been included for Figures 1–6 and Figure 5—figure supplement 1.

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
