## [Decision Letter]

**Acceptance summary:**

In this manuscript the authors have provided anatomical, physiological, and psychophysical evidence to better understand peripheral encoding of itch in human and non-human primate. In addition to demonstrating important differences in the mouse and human expression pattern of pruritogen receptor genes, their findings do not support the spatial contrast theory of itch vs pain generation, a theory that postulates that there is considerable overlap in the primary afferents that signal itch and pain.

**Decision letter after peer review:**

Thank you for submitting your article "Pruriception and neuronal coding in nociceptor subtypes in human and nonhuman primates" for consideration by *eLife*. Your article has been reviewed by 3 peer reviewers, and the evaluation has been overseen by a Reviewing Editor and Richard Aldrich as the Senior Editor. The following individuals involved in review of your submission have agreed to reveal their identity: Mahsa Sadeghi (Reviewer #1); H Richard Koerber (Reviewer #2).

The reviewers have discussed the reviews with one another and the Reviewing Editor has drafted this decision to help you prepare a revised submission.

We would like to draw your attention to changes in our policy on revisions we have made in response to COVID-19 (https://elifesciences.org/articles/57162). Specifically, when editors judge that a submitted work as a whole belongs in *eLife* but that some conclusions require a modest amount of additional new data, as they do with your paper, we are asking that the manuscript be revised to either limit claims to those supported by data in hand, or to explicitly state that the relevant conclusions require additional supporting data.

This manuscript describes a series of studies designed to investigate the coding of itch in primates. The authors combine findings from electrophysiological recordings from individual cutaneous fibers in non-human primates and the expression of pruriceptive receptors in human sensory neurons. The authors' findings challenge the spatial contrast theory for the coding of itch and pain.

Major concerns required additional experimental results for the revision: Two of the reviewers note that you did not examine the expression pattern of MrgprC and MrgprD in non-human primate DRG neurons. This is relevant as the expression pattern of mMrgA and mMrgC subfamilies differs considerably, even between different species of rodent. For example, colocalization of mMrgA and mMrgC subfamilies has been reported in mice, but not in rat.

Figure 6. You conclude that "the co-activation of different classes of CMHs or co-activation of CMHs and C-MIAs does not increase areas of dysesthesias, flare and/or wheal in comparison to the activation of distinct classes of nociceptors alone". Did you try lower doses of His alone, or in combination with AlA and BAM8-22? There is a possibility that the concentration of His or other pruritogen used reached the saturation level for inducing itch and pain. That would explain why there was no further increase in pain and itch sensation following administration of the three pruritogens together. If lower doses were indeed tested, then those data should be included in a revision. Otherwise, such experiments are critical to a correct interpretation of your results.

One reviewer suggests that calcium imaging of primate DRG neurons might address your concern regarding expression and functional data discrepancy, and certainly would help to address the functional activation of these receptors in the DRG. Although this would certainly be an excellent addition to the revision, at this time we do not consider this an essential experiment.

---

## [Author Response]

[…] Major concerns required additional experimental results for the revision: Two of the reviewers note that you did not examine the expression pattern of MrgprC and MrgprD in non-human primate DRG neurons. This is relevant as the expression pattern of mMrgA and mMrgC subfamilies differs considerably, even between different species of rodent. For example, colocalization of mMrgA and mMrgC subfamilies has been reported in mice, but not in rat.

The nomenclature and differences in genomic representation of the Mrgpr receptors in different species can be confusing and a little complicated. To be clear, the genomes of humans and non-human primates do not contain genes for any *Mrgpra* or *Mrgprc* receptor. Therefore, we interpret the Reviewers’ question as, “Did we examine the expression of MRGPRX1 and MRGPRD in non-human primates?” We answer this query by performing the later comparison in tissue from Macaca *nemestrina* DRG, because we could not examine the expression of Mrgpra and Mrgprc in monkey DRG. The new results from nonhuman primate DRG are illustrated in Figure 1—figure supplement 1, summarized in Supplementary File 1 and stated in lines 130-138, (pg. 67) as follows:

‘In parallel with these studies, we characterized the expression of MRGPRD and MRGPRX1 in DRG of macaques. […] Together our results point to profound species-dependent differences of MRGPR expression patterns between rodents and primates with additional slight differences inside the primate lineage.’

Further, in the section (lines 108-111, pg 6) of our manuscript where we introduce the species difference in Mrgpr, we amended the description and added a dendrogram (see Figure 1A), that schematically explains the species differences to help clarify the issue.

Figure 6. You conclude that "the co-activation of different classes of CMHs or co-activation of CMHs and C-MIAs does not increase areas of dysesthesias, flare and/or wheal in comparison to the activation of distinct classes of nociceptors alone". Did you try lower doses of His alone, or in combination with AlA and BAM8-22? There is a possibility that the concentration of His or other pruritogen used reached the saturation level for inducing itch and pain. That would explain why there was no further increase in pain and itch sensation following administration of the three pruritogens together. If lower doses were indeed tested, then those data should be included in a revision. Otherwise, such experiments are critical to a correct interpretation of your results.

In the psychophysical experiments our intent was not to test whether near threshold doses of different pruritogens (that partially activate fiber populations) are additive but rather to test whether combining suprathreshold doses (that effectively activate two or more fiber populations) increases the magnitude of itch vs. pain. The goal is to test whether a combination of pruritogens that activates more types of primary afferents and decreases the spatial contrast between activated and silent fibers, would increase pain and diminish itch. This point is now strengthened in the introduction (see lines 83-88, pg 4-5) and stated as follows:

‘However, it has not been tested in humans whether a combination of pruritogens that activates a greater number and variety of primary afferents and thereby decreases the spatial contrast between activated and silent fibers could lead to reduced itch and/or increased nociceptive sensation.’

As for testing lower doses of histamine, in a previous study (Simone at al., 1987), histamine was applied in human subjects over a range of 10^-4^-10^+2^ µg. While the flare area saturated at 10 µg, the sensation of itch increased throughout the dose range with the highest magnitude of itch being reported at 100 µg. Therefore, the histamine dose administered in the present study (10 µg) is not at the saturation dose for the sensation of itch.

One reviewer suggests that calcium imaging of primate DRG neurons might address your concern regarding expression and functional data discrepancy, and certainly would help to address the functional activation of these receptors in the DRG. Although this would certainly be an excellent addition to the revision, at this time we do not consider this an essential experiment.

In our manuscript (lines 552-558, pg 26) we have put forward several mechanisms that could account for the expression and functional data discrepancy. The responses to the different pruritogens may depend on where these are administered (peripheral cutaneous terminal vs soma). Therefore, calcium imaging experiments on neuronal DRG soma would not be able to resolve the discrepancy between RNA expression and the functional data observed after peripheral, cutaneous injection of the pruritogens. We agree with the reviewers that such experiments are not essential at this time.